# Alternative Analysis Approaches for the Assessment of Pilot Bioavailability/Bioequivalence Studies

**DOI:** 10.3390/pharmaceutics15051430

**Published:** 2023-05-07

**Authors:** Sara Carolina Henriques, João Albuquerque, Paulo Paixão, Luís Almeida, Nuno Elvas Silva

**Affiliations:** 1Research Institute for Medicines (iMed.ULisboa), Faculty of Pharmacy, Universidade de Lisboa, 1649-003 Lisboa, Portugal; ppaixao@ff.ulisboa.pt; 2BlueClinical Ltd., Senhora da Hora, 4460-439 Matosinhos, Portugal; joao.albuquerque@blueclinical.pt (J.A.); lalmeida@blueclinical.pt (L.A.); 3Centro de Estatística e Aplicações, Faculdade de Ciências, Universidade de Lisboa, 1749-016 Lisboa, Portugal

**Keywords:** bioequivalence, generic medicinal products, pilot studies, ƒ_2_ factor, bootstrap, pharmacokinetics, pharmacokinetic simulation

## Abstract

Pilot bioavailability/bioequivalence (BA/BE) studies are usually conducted and analysed similarly to pivotal studies. Their analysis and interpretation of results usually rely on the application of the average bioequivalence approach. However, due to the small study size, pilot studies are inarguably more sensitive to variability. The aim of this work is to propose alternative approaches to the average bioequivalence methodology, in a way to overcome and reduce the uncertainty on the conclusions of these studies and on the potential of test formulations. Several scenarios of pilot BA/BE crossover studies were simulated through population pharmacokinetic modelling. Each simulated BA/BE trial was analysed using the average bioequivalence approach. As alternative analyses, the centrality of the test-to-reference geometric least square means ratio (GMR), bootstrap bioequivalence analysis, and arithmetic (A_mean_) and geometric (G_mean_) mean ƒ_2_ factor approaches were investigated. Methods performance was measured with a confusion matrix. The G_mean_ ƒ_2_ factor using a cut-off of 35 was the most appropriate method in the simulation conditions frame, enabling to more accurately conclude the potential of test formulations, with a reduced sample size. For simplification, a decision tree is also proposed for appropriate planning of the sample size and subsequent analysis approach to be followed in pilot BA/BE trials.

## 1. Introduction

Bioavailability and bioequivalence are the cornerstone for the approval of brand-name and generic drugs globally under the European Medicines Agency (EMA) [1] and US Food and Drug Administration (FDA) [2] ambience. According to EMA, bioequivalence is the absence of a significant difference in the bioavailability (i.e., rate and extent) to which the active substance in pharmaceutical equivalents or pharmaceutical alternatives becomes available at the site of drug action when administered at the same molar dose, under similar conditions [1]. When bioequivalence between two drug products is claimed, an equivalent therapeutic efficacy and safety are assumed. Therefore, lengthy and costly phase III clinical trials on the bioequivalent test product may be waived [3].

A key goal in pharmaceutical development of oral dosage forms is a good understanding of the in vivo and in vitro performance of the dosage form and the optimization of an in vitro profile for the potential formulation that reflects its in vivo performance. In vitro dissolution testing provides useful information at several stages of the drug product development process and is usually used to assist scientists on excipients selection and manufacturing process adjustments that originate a candidate formulation with the most suitable and reproducible release profile. Therefore, dissolution results are commonly the decision key to test the new formulation in vivo. Nevertheless, dissolution results not always guarantee a correlation between the in vitro and the in vivo performance of the developed formulations [4]. Knowing this, and following a conservative approach, it is usual for companies to carry out pilot bioavailability/bioequivalence (BA/BE) studies.

A pilot BA/BE study is a downsized trial that can be conducted prior to the definitive pivotal trial and may act as a gatekeeping in vivo strategy to decide whether to move forward with a full-size pivotal study [4,5]. The pilot study can serve as a valuable tool (i) to validate the analytical methodology, (ii) to assess pharmacokinetic variability and to determine sample size to achieve adequate power, (iii) to optimize sample collection time intervals, (iv) to determine the needed washout period between treatments, and (v) to gather information about the formulation (or formulations) being tested against a reference product, and to assess its eligibility as a possible bioequivalent candidate(s) [1,2,5,6].

Pilot studies are usually conducted and analysed similarly to pivotal studies. Literature and guidelines provide no formal methodologies, besides the application of an average bioequivalence [4,6]. Average bioequivalence is a parametric approach based solely on the comparison of means, whether other characteristics of the distributions of the selected bioavailability metrics (e.g., inter- or intra-subject variabilities) are ignored. For formal pivotal BA/BE studies, test and reference formulations can be considered bioequivalent if the 90% confidence interval (CI) of the ratio of the geometric least square means of the pharmacokinetic parameters of interest are within the acceptance interval of [80.00–125.00]% [1,2,7].

The number of subjects to be included in a pilot study is generally 12–18 subjects, depending on the expected intra-subject coefficient of variation (ISCV%) [1,2]. However, due to their small sample size, pilot studies are inarguably more sensitive to variability. The point estimate obtained for the means ratio may not be close to the real population value, particularly when variability is high. Consequently, there is a greater risk of either (i) validating a bioinequivalent test formulation and proceeding further with a pivotal study or (ii) discarding a potentially bioequivalent formulation, by not conducting a pivotal study. Moreover, due to the small sample size, the 90% CI of the mean may be much wider and therefore fall out of the acceptance interval and reduce the probability of a positive decision.

The aim of this work is to propose alternative approaches to the average bioequivalence methodology that is generally applied to pilot studies, in a way to reduce the uncertainty on the conclusions of these studies and on the potential of test formulations. Several scenarios of pilot BA/BE crossover studies were simulated through population pharmacokinetic modelling, accounting for different inter-individual (IIV) and inter-occasion (IOV) levels of variability. Methods performance was measured with a confusion matrix.

## 2. Materials and Methods

A total of 32,000 BA/BE crossover trials (corresponding to 1,344,000 different concentration-time profiles) were simulated (i) accounting for different sample sizes, (ii) combining different IIV and/or IOV variability levels for the pharmacokinetic parameters, (iii) and considering no difference or a difference between test and reference products on the mean absorption rate constant (k_a_) (Figure 1).

Trial simulations and statistical analysis were performed with R version 4.0.3 (R Foundation for Scientific Computing, 2013).

### 2.1. Study Design

All studies were simulated as two-sequence (Sequence 1 and Sequence 2), two-treatment (test and reference), two-period crossover (2 × 2 × 2) studies, accounting for a range of 12–30 (in an increment of two) subjects. Subjects were randomized prior to pharmacokinetic simulation. A computer-generated balanced block-wise randomization list was appropriately generated according to the study sample size (Figure 1—Study Design).

### 2.2. Population Pharmacokinetic Simulation

For the simulation of plasma concentration-time profiles, a population pharmacokinetic modelling and simulation approach was used.

A one-compartmental model with first-order absorption and first-order elimination was selected as the simplest oral model to describe the processes of drug absorption and disposition. Simulations were performed through ordinary differential equations (ODE), parameterized with micro constants (Figure 1—Structural Model), and Equations (1) and (2), where A_GI_ represents the amount in the gastrointestinal tract, A_1_ the amount in the organism, k_a_ the absorption rate constant, k_e_ the elimination rate constant, V the apparent volume of distribution and C the plasma concentration.
(1)$$\left\{\;\begin{array}{c} \frac{{\mathrm{dA}}_{\mathrm{GI}}}{\mathrm{dt}}=-k_{a}\cdot A_{\mathrm{GI}}\\ \frac{{\mathrm{dA}}_{1}}{\mathrm{dt}}=k_{a}\cdot A_{\mathrm{GI}}-k_{e}\cdot A_{1} \end{array}\right.$$
(2)$$C=\frac{A_{1}}{V}$$

A range of 12–30 subjects per study was simulated using ‘Simulx’, a function of the ‘mlxR’ package version 4.1.3 (Monolix version 2019R2, Lixoft, Antony, France), that implements ODEs based mixed effects models by interfacing the C++ MlxLibrary with R. According to the previously defined sequence-balanced randomization scheme, subjects were administered a single 50 mg oral dose of either test or reference products, separated by a washout of 7 days (Figure 1—Study Design). Each pharmacokinetic profile comprised 20 simulated plasma samples, at the time of dose (time 0); and at 0.25, 0.50, 0.75, 1.00, 1.50, 1.75, 2.00, 2.25, 2.50, 2.75, 3.00, 3.25, 3.50, 3.75, 4.00, 6.00, 8.00, 12.00, and 24.00 h after dose. For each individual and occasion, compartmental pharmacokinetic parameters were generated considering one of the following six (6) different variability scenarios: (i) baseline (0% IIV and 0% IOV), (ii) 30% IIV and 0% IOV, (iii) 30% IIV and 10% IOV, (iv) 30% IIV and 20% IOV, (v) 30% IIV and 30% IOV, and (vi) 0% IIV and 45% IOV (Figure 1—Statistical Model). Each variability scenario was applied separately to each model parameter, i.e., variability scenarios were not applied simultaneously. Mean values for k_a_, V, and k_e_ are presented in Table 1. All parameters followed a log-normal distribution (Equation (3)). Absolute bioavailability (F) was considered to have a mean value of 0.9 (Table 1), and no variability was tested for this parameter.
(3)$$\log\left(\Psi_{i}\right)\;\sim \;N\left(\log\left({\bar{\Psi }}_{i}\right),\;\left(\omega^{2},\;\gamma^{2}\right)\right)\;\Leftrightarrow \;\Psi_{i}={\bar{\Psi }}_{i}\cdot e^{\eta_{i}+\kappa_{i}},\;\mathrm{where}\;\eta_{i}\;\sim \;N\left(0,\omega^{2}\right)\;\mathrm{and}\;\kappa_{i}\;\sim \;N\left(0,\gamma^{2}\right)$$

Furthermore, the individual plasma concentrations over time were simulated considering a log-normal additive experimental error (Equation (4)). A coefficient of variation (CV%) of 10% was used to reflect sampling and quantification errors. For simplicity, neither sequence nor period variability was included.
(4)$$Y=f\left(\theta ;x\right)\cdot e^{\varepsilon }\;\Leftrightarrow \;\log Y=\log\;f\left(\theta ;x\right)+\varepsilon$$

Moreover, for the purpose of this work, two groups of simulations were planned: one group where test and reference products were considered equal, with the same mean values for the pharmacokinetic model parameters (i.e., truly bioequivalent), and another group where test product presented a mean k_a_ value as 30% of the reference product mean k_a_ (i.e., truly bioinequivalent) (Figure 1—Covariate Model).

Within each group of simulations, and for each variability scenario, 100 bioequivalence crossover trials were simulated. The pharmacokinetic parameters maximum observed plasma concentration (C_max_) and area under the plasma concentration-time curve (AUC) were derived for each pharmacokinetic profile. The AUC typically reflects the extent of drug absorption, whether C_max_ is considered to reflect the absorption rate [1,2]. C_max_ usually shows larger variation compared to AUC, as the parameter highly depends on the selection of sampling times. Thus, as the risk of failing to demonstrate bioequivalence is higher for the rate of drug absorption, performed simulations only covered the effect of variability on the bioequivalence of C_max_.

### 2.3. Simulations Analysis

Each simulated bioequivalence trial was analysed using the average bioequivalence approach. As alternatives, the centrality of the test-to-reference GMR, a bootstrap bioequivalence analysis, and arithmetic (A_mean_) and geometric (G_mean_) mean ƒ_2_ factor approaches were also investigated.

#### 2.3.1. Average Bioequivalence Analysis

An analysis of variance (ANOVA) was performed on the *ln*-transformed C_max_. A linear model was applied, using sequence, subject nested within sequence, period and treatment as fixed effects [1,8,9].

As in accordance with EMA’s Guideline on the Investigation of Bioequivalence [1] and US FDA Guidance [2,7], for each simulated pilot study, the assessment of bioequivalence was based upon the 90% CI for the test-to-reference geometric least square means (LSM) ratio (GMR) for the primary pharmacokinetic parameter. This method is based on Schuirmann’s two one-sided *t*-tests (TOST) with the null hypothesis of bioinequivalence at the 5% significance level (α = 0.05) [1,2,3,7,10]. Assuming a maximum 20% difference between test and reference formulations, the interval hypotheses for average bioequivalence can be formulated as
(5)$$H_{0}:\;\mu_{T}-\mu_{R}<ln\left(0.80\right)\;\mathrm{or}\;\mu_{T}-\mu_{R}>-ln\left(0.80\right)\;\mathrm{versus}\;H_{1}:\;ln\left(0.80\right)\leq \mu_{T}-\mu_{R}\leq -ln\left(0.80\right)$$
where *μ*_T_ and *μ*_R_ are the population average response (i.e., the LSM) of the *ln*-transformed measure for test and reference formulations, respectively. Hence, for the back-transformed data, the hypotheses for average bioequivalence can be expressed as
(6)$$H_{0}:\;e^{\mu_{T}}/e^{\mu_{R}}<0.80\;\mathrm{or}\;e^{\mu_{T}}/e^{\mu_{R}}>1.25\;\mathrm{versus}\;H_{1}:\;0.80\leq e^{\mu_{T}}/e^{\mu_{R}}\leq 1.25$$
where the alternative hypothesis (H_1_) is shown by rejecting the null hypothesis (H_0_) of average bioinequivalence, i.e., the decision of bioequivalence is based on whether the 90% CI ($100\cdot \left(1-2\alpha \right)\%$) of the test-to-reference GMR is within the regulatory acceptance interval of [80.00–125.00]% [3,7,10,11].

Moreover, the ISCV% was estimated for each of the primary pharmacokinetic parameters, as
(7)$$\mathrm{ISCV}\%=100\;\cdot \sqrt{e^{s^{2}}-1}$$
where *s*^2^ is the mean square error obtained from the ANOVA model of the *ln*-transformed parameters [1,8,9].

Average bioequivalence analysis was performed through in-house functions developed in R, previously validated with Phoenix^®^ WinNonlin^®^ version 8.3 (Certara USA Inc., Princeton, NJ, USA).

#### 2.3.2. Centrality of the Test-to-Reference GMR

Beyond the standard average bioequivalence approach, the centrality of the test-to-reference GMR was tested, i.e., for each simulated pilot study it was verified if the attained test-to-reference GMR was within the tighter acceptance range of [90.00–111.11]%.

#### 2.3.3. Bootstrap Bioequivalence Analysis

Bootstrapping is a non-parametric method that can be used to assess the precision of a statistic without making strong assumption for the distribution from which samples are drawn [12].

Using Monte Carlo simulations, new sets of pharmacokinetic data were created, by repeatedly sampling from the simulated study data with replacement. By resampling with replacement, the bootstrap resampling mimics the experimental procedure [12].

For each simulated study, a total of 100 bootstrap resamples were generated from the simulated concentration-time profiles [12,13]. The sample size of the bootstrap resamples was calculated based on the original simulated study, from which the average bioequivalence approach was applied and ISCV% was estimated. For sample size calculation for bootstrap resampling, it was assumed the estimated ISCV%, a power of 80%, a true test-to-reference GMR of 90% and an α of 0.05, using R package ‘PowerTOST’ version 1.5–3 [14]. To ensure that the number of resampled subjects assigned with sequences 1 and 2 was the same, resampling was sequence balanced.

From the resampled data set, average bioequivalence was re-computed, thus generating a bootstrap estimate of the statistics of interest. The bootstrap resamples’ GMR were then used to estimate the standard error of the bootstrap GMR and its corresponding 95% CI. The non-parametric confidence bounds were obtained as percentiles from the bootstrap estimator of the sampling distribution of the parameter estimator [12,15]. A was reduced to 0.025, in order to better circumvent type I errors. Similarly, the decision of bioequivalence was based on whether the bootstrap 95% CI of the test-to-reference GMR was within the regulatory acceptance interval of [80.00–125.00]%.

#### 2.3.4. Similarity ƒ_2_ Factor

The similarity ƒ_2_ factor is a mathematical index widely used to compare dissolution profiles, evaluating their similarity, using the percentage of drug dissolved per unit of time. The similarity ƒ_2_ factor, proposed by Moore and Flanner in 1996 [16], is derived from the mean squared difference, and can be calculated as a function of the reciprocal of mean squared-root transformation of the sum of square differences at all points:(8)$$f_{2}=50\;\cdot \log\left(100\;\cdot {\left[1+\frac{1}{n}\overset{t=n}{\underset{t=1}{\sum }}{\left({\bar{R}}_{t}-{\bar{T}}_{t}\right)}^{2}\right]}^{-0.5}\right)$$
where ƒ_2_ is the similarity factor, *n* is the number of time points, and ${\bar{R}}_{t}$ and ${\bar{T}}_{t}$ are the mean percentage of drug dissolved at time *t* after initiation of the study, for reference and test products, respectively [1,16,17].

The ƒ_2_ similarity factor ranges from 0 (when ${\bar{R}}_{t}-{\bar{T}}_{t}=100\%$, at all *t*) to 100 (when ${\bar{R}}_{t}-{\bar{T}}_{t}=0\%$, at all *t*). Therefore, applying Equation (8), an average difference of 10%, 15%, and 20% from all measured time points results in a ƒ_2_ value of 50, 41, and 35, respectively (Figure 2). EMA [1] and FDA [18,19] have set a public standard of ƒ_2_ value between 50–100, i.e., a maximum mean difference of 10%, to indicate similarity between the two dissolution profiles.

In this work, the concept of similarity factor ƒ_2_ was applied as an alternative to the average bioequivalence analysis. The similarity between test and reference products by means of ƒ_2_ was evaluated through the comparison of arithmetic (A_mean_) and geometric (G_mean_) means of plasma concentration-time profiles derived from the simulated individual pharmacokinetic profiles. ƒ_2_ was used to assess the similarity on the rate of drug absorption by normalizing test and reference mean concentration-time profiles to the maximum plasma concentration (C_max_) derived from the mean reference profile, until reference C_max_ is observed (reference t_max_) (Equation (9)).
(9)$$C_{t}^{N}=100\cdot \frac{{\bar{C}}_{t}}{C_{\max}{}_{R}},\;\;\mathrm{where}\;0\leq t\leq t_{\max}{}_{R}$$

In Equation (9), $C_{t}^{N}$ is the normalized concentration at time *t*, ${\bar{C}}_{t}$ is the mean (test or reference) concentration at time *t*, $C_{\max}{}_{R}$ is the C_max_ of the reference mean concentration-time profile, and $t_{\max}{}_{R}$ the time of observation of $C_{\max}{}_{R}$. The similarity ƒ_2_ factor is calculated as
(10)$$C_{\max}\;f_{2}=50\cdot \log\left(100\cdot {\left[1+\frac{1}{n}\overset{t=n}{\underset{t=1}{\sum }}{\left(R_{t}^{N}-T_{t}^{N}\right)}^{2}\right]}^{-0.5}\right)$$
where *n* is the number of time points until reference t_max_, and $R_{t}^{N}$ and $T_{t}^{N}$ are the normalized concentration at time t for reference and test products, respectively.

Then, the ƒ_2_ factor was tested for differences between test and reference formulations’ mean concentration-time profiles of 10%, 15% and 20%. Consequently, the interval hypotheses for the ƒ_2_ factor can be formulated as
(11)$$H_{0}:\;f_{2}<\theta \;\mathrm{versus}\;H_{1}:\;f_{2}\geq \theta .$$
where θ is the tested cut-off of (i) 35 for testing maximum differences of 20%; (ii) 41 for differences of 15%; and (iii) 50 for testing differences of 10% between the concentration-time profiles.

### 2.4. Performance Measurement

When testing a hypothesis, two errors may occur: (i) the type I error, which concerns the rejection of true H_0_ (Equation (12)); (ii) and the type II error which concerns the failing to reject false H_0_ (Equation (13)) [3]. The probabilities of making type I and type II errors are given as
(12)$$\alpha =P\left(\mathrm{type}\;I\;\mathrm{error}\right)=P\left({\mathrm{reject}\;H}_{0}\;|{\;H}_{0}\;\mathrm{is}\;\mathrm{true}\right)$$
(13)$$\beta =P\left(\mathrm{type}\;\mathrm{II}\;\mathrm{error}\right)=P\left({\mathrm{fail}\;\mathrm{to}\;\mathrm{reject}\;H}_{0}\;|{\;H}_{0}\;\mathrm{is}\;\mathrm{false}\right)$$

In order to find the relationship between type I and type II errors, hence determining the performance of each bioequivalence evaluation method (average bioequivalence, centrality of the test-to-reference GMR, bootstrap bioequivalence, and A_mean_ and G_mean_ ƒ_2_ factor evaluated with a cut-off of 35, 41, and 50), a confusion matrix, i.e., a cross-tabulation of the observed and predicted classes with associated statistics, was created (Table 2).

For each evaluation method, the created matrixes accommodated (i) the true positives (TP), i.e., the number of correctly identified bioequivalent predictions; (ii) the false negatives (FN), i.e., the number of incorrectly identified bioinequivalent predictions; (iii) the false positives (FP), i.e., the number of incorrectly identified bioequivalent predictions; and (iv) the true negatives (TN), i.e., the number of correctly identified bioinequivalent predictions (Table 2).

Moreover, the following statistics were derived from the cross-tabulated matrix [20]:
Sensitivity, also referred to as power, recall or true positive rate, which measures the capacity of the model to correctly identify bioequivalent test and reference formulations. In other words, it is the probability of correctly rejecting H_0_ when H_0_ is false (Table 2).
(14)$$\mathrm{Sensitivity}\;or\;\mathrm{Power}=1-\beta =P({\mathrm{reject}\;H}_{0}\;|{\;H}_{0}\;\mathrm{is}\;\mathrm{false})$$When the test recognizes all the bioequivalent formulations (i.e., no false negatives) Sensitivity = 1; when the test does not recognize any of the bioequivalent formulations Sensitivity = 0.Specificity, also referred to as true negative rate, measures the capacity of the model to correctly identify bioinequivalent test and reference formulations. In other words, it is the probability of correctly failing to reject H_0_ when H_0_ is true (Table 2).
(15)$$\mathrm{Specificity}=1-\alpha =P({\mathrm{fail}\;\mathrm{to}\;\mathrm{reject}\;H}_{0}\;|{\;H}_{0}\;\mathrm{is}\;\mathrm{true})$$When the test recognizes all the bioinequivalent formulations (i.e., no false positives) Specificity = 1; when the test does not recognize any of the bioinequivalent formulations Specificity = 0.Precision, also referred to as positive predictive value (PPV), measures the correctness achieved in bioequivalent predictions (Table 2).When PPV = 1, all identified bioequivalent formulations are truly bioequivalent.Negative Predictive Value (NPV), which measures the correctness achieved in bioinequivalent predictions (Table 2).When NPV = 1, all identified bioinequivalent formulations are truly bioinequivalent.Accuracy, which represents the ratio between the correctly identified predicted instances (bioequivalent and bioinequivalent) and the total number of instances (Table 2).When Accuracy = 1, the test predicted correctly all the bioequivalent and bioinequivalent formulations.F_1_ score, which is the harmonic mean of Sensitivity and Precision.
(16)$$F_{1}=\frac{2\cdot \mathrm{Sensitivity}\cdot \mathrm{Precision}}{\mathrm{Sensitivity}+\mathrm{Precision}}$$F_1_ score is independent from the number of samples correctly classified as negative. A F_1_ = 1 indicates perfect precision and sensitivity; for a F_1_ = 0, either precision or sensitivity are 0.Matthews Correlation Coefficient (MCC), which measures the correlation coefficient between the true classes and the method predicted classes.
(17)$$\mathrm{MCC}=\frac{Cov\left(t,p\right)}{\sigma_{t}\cdot \sigma_{p}}=\frac{\mathrm{TP}\cdot \mathrm{TN}-\mathrm{FP}\cdot \mathrm{FN}}{\sqrt{\left(\mathrm{TP}+\mathrm{FP}\right)\cdot \left(\mathrm{TP}+\mathrm{FN}\right)\cdot \left(\mathrm{TN}+\mathrm{FP}\right)\cdot \left(\mathrm{TN}+\mathrm{FN}\right)}}$$
where *Cov*(*t*, *p*) is the covariance of the true classes *t* and predicted labels *p*, whereas *σ_t_* and *σ_p_* are the standard deviations, respectively [21]. A MCC = 1 indicates a perfect prediction; MCC = 0 indicates that the prediction is no better than random; and MCC = −1 indicates total disagreement between prediction and observation.Cohen’s Kappa (κ) statistic, which is a measure of concordance for categorical data that measures agreement relative to what would be expected by chance.
(18)$$\kappa =\frac{2\cdot \left(\mathrm{TP}\cdot \mathrm{TN}-\mathrm{FP}\cdot \mathrm{FN}\right)}{\left(\mathrm{TP}+\mathrm{FP}\right)\cdot \left(\mathrm{TP}+\mathrm{FN}\right)\cdot \left(\mathrm{TN}+\mathrm{FP}\right)\cdot \left(\mathrm{TN}+\mathrm{FN}\right)}$$When there is complete agreement κ = 1; when there is no agreement κ = 0; and when there is no effective agreement, or when there is a complete disagreement, κ = −1.

## 3. Results

### 3.1. Simulated Pharmacokinetic Data

Histograms of the individual estimates of pharmacokinetic parameters exhibited centred distribution around the population’s typical value. Descriptive statistics of all simulated pharmacokinetic parameters and a graphical representation of their distribution are presented in Appendix A.

The defined sampling times were appropriate to describe the simulated concentration-time profiles (Appendix A).

For the group of simulations concerning the test product as truly bioequivalent to the reference product, it was observed for the baseline simulations (0% IIV and 0% IOV) that the C_max_ for both test and reference products was approximately 642.5 µg/L (geometric coefficient of variation [GCV%] ≈ 6%), being reached between 0.75 and 4 h (median t_max_ = 2.25 h). AUC from pre-dose until the last sampling time was approximately 4950 µg.h/L (GCV% ≈ 3%). As expected, no differences between test and reference products were observed for these NCA parameters (Appendix A).

For the group of simulations concerning the test product as truly bioinequivalent to the reference product, it was observed a delayed t_max_ for the test product (t_max_ = 3.5 h [1.75–8 h]), as well as a 30% reduction of C_max_ (G_mean_ = 460.69 µg/L [GCV% ≈ 6%]), as a consequence of the differences in k_a_ between test and reference products. No differences were observed for AUC, as test and reference products only presented differences in k_a_, and not in F (Appendix A).

The increment in the variability of k_a_ (IIV and IOV) did not greatly affect the distribution of C_max_ nor AUC values (GCV% ≈ 7–20%); however, it increased the time range for the observation of C_max_. t_max_ values ranged from 0.5 to 6 h for the reference product and ranged from 0.25 to 8 h for a truly bioequivalent test, and from 1 to 12 h for a truly bioinequivalent test (Appendix A).

Likewise, an increase of variability in k_e_ was not associated with higher dispersion of C_max_ values, but it was associated with a wider time range for t_max_ (from 0.5 to 8 h for reference and truly bioequivalent test; and from 1 to 12 h for truly bioinequivalent test). An increase variability in k_e_ induced an increased variability in AUC (GCV% ≈ 27–40%) (Appendix A).

Moreover, an increase of variability in V was associated with a wider dispersion of C_max_ and AUC values (GCV% ≈ 30–48%). However, no differences were observed for t_max_ range (Appendix A).

### 3.2. Bioequivalence Evaluation

Confusion matrix results were graphically presented over the number of subjects for sensitivity in Figure 3 and Figure 4, for specificity in Figure 5, for precision in Figure 6, for NPV in Figure 7, for accuracy in Figure 8, for F_1_ in Figure 9, for MCC in Figure 10 and for Cohen’s κ in Figure 11.

Considering the evaluation of bioequivalence, as expected, the presence of type I and type II errors depended only on IOV, and not on IIV, as IIV is suppressed by using a crossover design.

For the pharmacokinetic model tested, IOV in V was the variability identified with the highest impact on the evaluation of C_max_ bioequivalence metric. The variability tested for the other model parameters had no relevant impact.

Variability (IOV) in V turns critical for bioequivalence evaluation at 20% IOV, a level of variability lower than the established cut-off value of 30% that defines a highly variable drug/drug product, in accordance with EMA’s Guideline [1].

#### 3.2.1. Average Bioequivalence Method

For an IOV of 20% in V and based on the sensitivity evaluation in studies with 12 subjects, the average bioequivalence method correctly identified 56% of the truly bioequivalent test formulations. When incrementing the sample size for 30 subjects, the sensitivity of the average bioequivalence method increased to 99%. When IOV increased to 30%, the sensitivity of the average bioequivalence method decreased to 15% and 76% for studies with 12 and 30 subjects, respectively. For an IOV of 45%, the sensitivity decreased to 1.0% and 21% for studies with 12 and 30 subjects, respectively (Table 3, Figure 3) Nevertheless, the average bioequivalence method performed well avoiding type I errors, even on the highest level of variability, with a specificity rounding the 100% (Table 3, Figure 5).

#### 3.2.2. Centrality of the Test-to-Reference GMR Method

For the simulated scenarios, the centrality of the point estimate (within [90.00–111.11]%) derived from the average bioequivalence approach showed a higher sensitivity than the corresponding 90% CI (Table 4, Figure 3). For a 20% IOV in V, this method correctly identified 79% and 99% of the truly bioequivalent test formulations to be within [90.00–111.11]% in studies with 12 subjects and with 30 subjects, respectively. When increasing IOV to 30%, the sensitivity of the method decreased to 57% and 85% in studies with 12 and 30 subjects, respectively. For an IOV of 45%, the sensitivity of the method decreased to 36% and 54% in studies with 12 and 30 subjects, respectively (Table 4, Figure 3). In terms of specificity, for an IOV of 45%, the centrality of the point estimate method led to an inflation of type I error to 10% in studies simulated with 12 subjects, which can be minimized by increasing the number of subjects in the trials (Table 4, Figure 5).

#### 3.2.3. Bootstrap Bioequivalence Method

The bootstrap bioequivalence method showed a higher sensitivity than the standard parametric approach. For an IOV of 20% in V, this non-parametric method correctly identified more than 90% of truly bioequivalent formulations, irrespective of sample size. When increasing IOV to 30%, the sensitivity decreased to 76% in studies with 12 subjects but scored higher than 90% in studies with 30 subjects (Table 5, Figure 3). For the highest tested variability (IOV of 45%), bootstrap sensitivity was 62% in studies with 12 subjects and 66% in studies with 30 subjects. On the other hand, bootstrap was the method that induced most type I errors (Table 5, Figure 5).

#### 3.2.4. Similarity *f*_2_ Factor Method

The ƒ_2_ factor derived from the A_mean_ and G_mean_ pharmacokinetic profiles behaved similarly. For an IOV of 20% in V, and using a cut-off of 35 (i.e., to detect a mean difference of 20%), the ƒ_2_ method could correctly identify more than 99% of truly bioequivalent test formulations in studies with 12 and 30 subjects. When increasing IOV to 30%, the sensitivity slightly decreased to 94% in studies with 12 subjects but scored higher than 98% in studies with 30 subjects. For the highest tested variability (IOV of 45%), the ƒ_2_ factor derived from both A_mean_ and G_mean_ profiles was found to be a much more sensitive approach than the standard average bioequivalence approach, with >76% and 96% of truly bioequivalent test formulations identified in studies with 12 and 30 subjects, respectively (Table 6 and Table 7, and Figure 3).

Using a cut-off of 41 (i.e., to detect a mean difference of 15%), the ƒ_2_ factor method still performed better than the average bioavailability method. As expected, the sensitivity slightly decreased while using a higher cut-off, however, differences in the sensitivity between 35 and 41 cut-off values were only noticeable at 30% IOV. For an IOV of 30% in V, the ƒ_2_ method could correctly identify more than 84% of truly bioequivalent test formulations with 12 subjects (nearly a 10% decrease in comparison to a cut-off of 35) and 98% with 30 subjects (no difference between the two cut-off values). Moreover, for the highest tested variability (IOV of 45%), the sensitivity of ƒ_2_ factor method for both A_mean_ and G_mean_ profiles using a cut-off of 41 decreased to nearly 66% and 88% (nearly 10% decrease in comparison to a cut-off of 35) with 12 and 30 subjects, respectively (Table 8 and Table 9, and Figure 3).

Using a cut-off of 50 (i.e., to detect a mean difference of 10%), ƒ_2_ factor method performed slightly worse than the average bioequivalence method in studies simulated for the lowest sample size (12 subjects) with the highest variability (IOV of 45%) on k_a_. In this case, a sensitivity of 88% was attained. However, regarding the different variability scenarios in V, the ƒ_2_ factor method using a cut-off of 50 was always more sensitive than the average bioequivalence method for an IOV ≥ 20%. Nevertheless, as expected, the sensitivity decreased, compared to the other tested cut-offs. For an IOV of 20% in V and using a cut-off value of 50, ƒ_2_ factor method correctly predicted nearly 80% of the truly bioequivalent test formulations with only 12 subjects and 99% with 30 subjects. For an IOV of 30%, ƒ_2_ factor method showed a sensitivity of more than 60% in studies with 12 subjects and more than 90% with 30 subjects. For the highest tested variability (IOV of 45%), ƒ_2_ factor correctly predicted almost 50% of the truly bioequivalent test formulations with 12 subjects and more than 64% with 30 subjects (Table 10 and Table 11, and Figure 3).

Along with the higher sensitivity shown by the ƒ_2_ factor method using different cut-offs in comparison to the average bioequivalence method, no inflation of type I error (>5%) was induced with ƒ_2_ factor method for A_mean_ and G_mean_ pharmacokinetic profiles, using all cut-off values (Table 5, Table 6, Table 7, Table 8, Table 9, Table 10 and Table 11, Figure 5).

#### 3.2.5. Comparison of Average Bioequivalence, Centrality of the Point Estimate, Bootstrap Bioequivalence, and Similarity ƒ_2_ Factor Methods

Accuracy, MCC, F_1_ and κ were calculated in order to select the best methodology to assess the potential of a test formulation to be bioequivalent to a reference formulation on the rate of drug absorption, based on pilot BA/BE trials.

In general, average bioequivalence was the least accurate approach. For an IOV of 20% in V and for a minimum sample size of 12 subjects, the average bioequivalence method showed an accuracy of 78% (Table 3), while the other approaches scored ≥89.5% (Figure 8). When increasing the sample size to 30 subjects, all methods were ≥99% accurate. By increasing IOV to 30% in V, the accuracy of the average bioequivalence approach decreased to 57% and 88% with 12 and 30 subjects, respectively (Table 3); and the accuracy of the centrality of the GMR approach scored between 76.5 and 92.5% with 12 and 30 subjects, respectively (Table 4). All the other methods scored similarly, with an accuracy above 80% for studies with 12 subjects and above 95% for studies with 30 subjects (Figure 8). At the highest tested level of variability (IOV of 45%), the accuracy of the average bioequivalence method decreased to 50.5% and 60.5% with sample sizes of 12 and 30 subjects, respectively (Table 3); and the accuracy of the centrality of the point estimate method was decreased to 63% and 76% for studies with 12 and 30 subjects, respectively (Table 4). The bootstrap bioequivalence method showed an accuracy of 72% and 80.5% in studies with 12 and 30 subjects, respectively (Table 5). Regarding the ƒ_2_ factor method derived from A_mean_ and G_mean_ pharmacokinetic profiles, the accuracy was above 80% and 94% in studies with 12 and 30 subjects, respectively, using a cut-off of 35 (Table 6 and Table 7) and a cut-off of 41 (Table 8 and Table 9); while using a cut-off of 50, accuracy ranged from 74% to 82% in studies with 12 and 30 subjects, respectively (Table 10 and Table 11).

Similarly, average bioequivalence was the method with the lowest harmonic mean between sensitivity and precision (F_1_) (Figure 9). For an IOV of 20% in V, and for studies with 12 and 30 subjects, F_1_ estimates for the average bioequivalence ranged between 71.8% and 99.5%, respectively (Table 3); for the centrality of the GMR ranged between 88.3% and 99.5% (Table 4); and for bootstrap bioequivalence ranged between 94.3% and 99.5% (Table 5). For the ƒ_2_ factor derived from A_mean_ and G_mean_ pharmacokinetic profiles, F_1_ was above 99% using a cut-off of 35 (Table 6 and Table 7), above 98% using a cut-off and 41 (Table 8 and Table 9), while using a cut-off of 50, F_1_ ranged from 88% to 99% in studies with 12 and 30 subjects, respectively (Table 10 and Table 11).

For an IOV of 30% and for studies with 12 and 30 subjects, average bioequivalence F_1_ highly decreased, ranging between 25.9% and 86.4%, respectively (Table 3). For the same sample sizes, the centrality of the GMR method showed an F_1_ between 70.8% and 91.9% (Table 4), respectively, and the bootstrap bioequivalence method presented an F_1_ between 83.5% and 96.4% (Table 5). For the ƒ_2_ factor derived from A_mean_ and G_mean_ pharmacokinetic profiles, F_1_ was above 96% using a cut-off of 35 (Table 6 and Table 7), and above 91% using a cut-off of 41 (Table 8 and Table 9). Using a cut-off of 50, F_1_ ranged from 75% to 95% in studies with 12 and 30 subjects, respectively (Table 10 and Table 11).

For the highest IOV (45%) in V, and for studies with 12 and 30 subjects, average bioequivalence F_1_ decreased to 2% and 34.7%, respectively (Table 3). For the same sample sizes, the centrality of the GMR showed an F_1_ between 49.3% and 69.2% (Table 4), respectively, and the bootstrap bioequivalence method presented an F_1_ of 68.9% and 77.2% (Table 5). For the ƒ_2_ factor derived from A_mean_ and G_mean_ pharmacokinetic profiles, F_1_ ranged within 85% and 98% using a cut-off of 35 (Table 6 and Table 7), within 80% and 94% using a cut-off and 41 (Table 8 and Table 9); while using a cut-off of 50, F_1_ ranged from 65% to 78% in studies with 12 and 30 subjects, respectively (Table 10 and Table 11).

Considering the correlation between the true classes and the predicted labels (MCC) (Figure 10), average bioequivalence was the method that scored lower. For an IOV of 20% in V, average bioequivalence MCC ranged between 62.4% and 99% in studies with 12 and 30 subjects, respectively (Table 3). For the same sample sizes, MCC for the centrality of the point estimate ranged between 80.8% and 99.0% (Table 4) and MCC for the bootstrap bioequivalence ranged between 89.2% and 99% (Table 5). For the ƒ_2_ factor derived from A_mean_ and G_mean_ pharmacokinetic profiles, MCC ranged within 99% and 100% using a cut-off of 35 (Table 6 and Table 7), within 97% and 100% using a cut-off and 41 (Table 8 and Table 9); while using a cut-off of 50, MCC ranged from 80% to 98% in studies with 12 and 30 subjects, respectively (Table 10 and Table 11).

For an IOV of 30%, average bioequivalence MCC ranged between 25.8% and 78.3% in studies with 12 and 30 subjects, respectively (Table 3). For the same sample sizes, MCC for the centrality of the point estimate ranged between 57.6% and 86.0% (Table 4) and MCC for the bootstrap bioequivalence ranged between 71.2% and 93.1% (Table 5). For the ƒ_2_ factor derived from A_mean_ and G_mean_ pharmacokinetic profiles, MCC ranged within 94% and 100% using a cut-off of 35 (Table 6 and Table 7), within 85% and 98% using a cut-off and 41 (Table 8 and Table 9); while using a cut-off of 50, MCC ranged from 66% to 91% in studies with 12 and 30 subjects, respectively (Table 10 and Table 11).

For the highest IOV (45%) in V, average bioequivalence MCC decreased to 7.10% and 34.3% in studies with 12 and 30 subjects, respectively (Table 3). For the same sample sizes, MCC for the centrality of the point estimate ranged between 30.9% and 57.9% (Table 4) and MCC for the bootstrap bioequivalence ranged between 44.9% and 63.7% (Table 5). For the ƒ_2_ factor derived from A_mean_ and G_mean_ pharmacokinetic profiles, MCC ranged between 77% and 96% using a cut-off of 35 (Table 6 and Table 7), within 70% and 89% using a cut-off and 41 (Table 8 and Table 9); while using a cut-off of 50, MCC ranged from 56% to 70% in studies with 12 and 30 subjects, respectively (Table 10 and Table 11).

Average bioequivalence was the method with the lowest concordance agreement relative to what would be expected by chance (κ) (Figure 11). For an IOV of 20% in V, average bioequivalence κ ranged between 56% and 99% in studies with 12 and 30 subjects, respectively (Table 3); For the same sample sizes, κ for the centrality of the point estimate ranged between 79.0% and 100% (Table 4) and κ for bootstrap bioequivalence ranged between 89.0% and 99.0% (Table 5). For the ƒ_2_ factor derived from A_mean_ and G_mean_ pharmacokinetic profiles, κ was above 99% using a cut-off of 35 (Table 6 and Table 7), above 97% using a cut-off and 41 (Table 8 and Table 9); while using a cut-off of 50, κ ranged from 79% to 99% in studies with 12 and 30 subjects, respectively (Table 10 and Table 11).

For an IOV of 30%, average bioequivalence κ ranged between 14% and 76% in studies with 12 and 30 subjects, respectively (Table 3). For the same sample sizes, the centrality of the point estimate ranged between 53% and 85% (Table 4) and κ for the bootstrap bioequivalence ranged between 70% and 93% (Table 5). For the ƒ_2_ factor derived from A_mean_ and G_mean_ pharmacokinetic profiles, κ was above 94% using a cut-off of 35 (Table 6 and Table 7), above 84% using a cut-off and 41 (Table 8 and Table 9); while using a cut-off of 50, κ ranged from 61% to 92% in studies with 12 and 30 subjects, respectively (Table 10 and Table 11).

For the highest IOV (45%) in V, average bioequivalence κ decreased to 1% and 21% in studies with 12 and 30 subjects, respectively (Table 3). For the same sample sizes, κ for the centrality of the point estimate ranged between 26.0% and 52.0% (Table 4) and κ for bootstrap bioequivalence ranged between 44% and 61% (Table 5). For the ƒ_2_ factor derived from A_mean_ and G_mean_ pharmacokinetic profiles, κ was within 75% and 96% using a cut-off of 35 (Table 6 and Table 7), within 66% and 89% using a cut-off and 41 (Table 8 and Table 9); while using a cut-off of 50, κ ranged from 48% to 66% in studies with 12 and 30 subjects, respectively (Table 10 and Table 11).

## 4. Discussion

For the pharmacokinetic model tested, an increment in the variability of V was associated with a higher dispersion of C_max_ values (GCV% ≈ 30–48%). Hence, the within-individual variability (IOV) in V was the identified variability with the highest impact on the bioequivalence evaluation of C_max_.

For each bioequivalence evaluation method (average bioequivalence, centrality of the test-to-reference GMR, bootstrap bioequivalence, and A_mean_ and G_mean_ ƒ_2_ factor evaluated with a cut-off of 35, 41, and 50) the relationship between type I and type II errors was studied. Moreover, accuracy, MCC, F_1_, and κ were calculated in order to select the best methodology for the evaluation of the potentiality of a test formulation to be bioequivalent to a reference formulation on the rate of drug absorption, based on pilot BA/BE trials. For each bioequivalence evaluation method, results were consistent for all the calculated cross-tabulation matrix statistics.

Average bioequivalence was found to be the most underpower method tested, i.e., that induced higher type II errors (Table 3 and Figure 3). A critical decrease in sensitivity was observed for an IOV of 20% (in V), a level of variability lower than the cut-off value of 30% established by the EMA’s Guideline on the Evaluation of Bioequivalence for highly variable drug/drug products [1]. Similarly, average bioequivalence was the method that scored lower for all the other performance measures. Nevertheless, inflation of the type I error was not observed for this statistical method, being kept below 0.05 (Table 3 and Figure 5).

The newly proposed approaches (centrality of the test-to-reference GMR, bootstrap bioequivalence, and A_mean_ and G_mean_ ƒ_2_ factor) showed a higher sensitivity/power than the established average bioequivalence method commonly used (Figure 3).

The alternative methodologies can maintain a power of at least 80% with less than 20 subjects in studies with a high IOV (30%), while the average bioequivalence approach required at least 80 subjects to maintain the same power level (Figure 3, Table 12).

Moreover, the alternative methods showed a higher performance for the other cross-tabulated matrix statistics, i.e., a better concordance between the truth and predictions. Hence, for downsized trials as pilot studies, the use of the proposed alternative approaches may reduce the uncertainty in the evaluation of the potentiality of a test formulation to be bioequivalent to a reference formulation on the rate of drug absorption, helping pharmaceutical companies on the decision to go forward to pivotal bioequivalence studies.

Regarding the centrality of the GMR, the method showed a higher sensitivity than the average bioequivalence method. However, using this alternative can be misleading, as it may lead to false positives due to its lower specificity for higher variabilities.

The bootstrap methodology was able to maintain a power of at least 80% for simulations with an IOV of 20% in V and a sample size of 12 subjects (Table 5 and Figure 3), as well as for simulations with an IOV of 30% and a sample size of only 14 subjects (Table 12). These sample sizes correspond to 75% and 44% of the sample size estimated based on the same assumptions of IOV (20% and 30%) and expected power level (80%), using the average bioequivalence analysis approach (i.e., 16 and 32 subjects, respectively, Table 12). However, this non-parametric approach was the method that induced higher type I rates (Table 5 and Figure 5). Nevertheless, the bootstrap bioequivalence method was found to be more accurate than the standard average bioequivalence.

Additionally, ƒ_2_ factor methodology was tested using cut-offs of 35, 41, and 50 for testing a mean difference of 20%, 15%, and 10%, respectively, between the concentration-time profiles of test and reference, until the reference C_max_.

Regarding the ƒ_2_ factor methodology, for an IOV of 20% (in V), 12 subjects are needed to target a power of at least 80%, either using a cut-off of 35 or 41, corresponding to 75% of the required sample size estimated with the same IOV and power assumptions, using the average bioequivalence analysis approach (Table 12). Using a cut-off of 50, 14 subjects would be needed (corresponding to 88% of the estimated sample size using the average bioequivalence analysis approach). For an IOV of 30%, and to target a power of at least 80%, 12 subjects are necessary using a cut-off of 35 and 41 (corresponding to 38% of the estimated sample size using the average bioequivalence analysis approach). For the highest tested variability (45%) and to target the same power level of at least 80%, pilot studies may be performed with 14 subjects (using a cut-off of 35) or 20 subjects (using a cut-off of 41), which correspond to 21% and 30%, respectively, of the estimated sample size using the average bioequivalence analysis approach.

Moreover, considering that none of the tested ƒ_2_ factor cut-offs inflated type I error rate (a maximum type I error of only 1% was observed for A_mean_ ƒ_2_ factor with a cut-off of 35 for simulations performed using an IOV of 45% in V [Table 6]), the authors suggest the use of a cut-off of 35 instead of 41 and 50 for the ƒ_2_ factor methodology, under the simulated conditions frame.

Despite minor differences were observed for ƒ_2_ factor derived from the A_mean_ and G_mean_ pharmacokinetic profiles, the G_mean_ ƒ_2_ factor using a cut-off of 35 was the method with the best relationship between avoiding type I and type II errors. It was also the method with higher accuracy and a better relationship between outcomes and predictions. Nevertheless, simulations are needed with more extreme scenarios (e.g., a true GMR of 90% and 80%) to better define a cut-off for this method.

A correlation between G_mean_ ƒ_2_ factor and GMR and the absolute true mean difference of *ln*-transformed test and reference C_max_ (i.e., LSM) is shown in Figure 12. The higher the absolute true mean difference of *ln*-transformed test and reference C_max_, the lower the ƒ_2_ factor. Moreover, this figure also shows that more accurate GMR and ƒ_2_ factor estimates are obtained with the increase of the number of simulated subjects in the trial (for true bioequivalent simulations, GMR is 100% and ƒ_2_ factor is 70; for true bioinequivalent simulations, GMR is 70% and ƒ_2_ factor is 20).

Based on the results of this work, the authors propose in Figure 13 a decision tree with a rationale for sample size and analysis approach to be followed when planning a pilot BA/BE trial for drugs characterized by a median t_max_ of approximately 2 to 4 h.

## 5. Conclusions

Given the uncertainty of results derived from pilot BA/BE trials performed with drug/drug products showing a considerable variability (IOV > 20%), and consequently the uncertainty on the conclusions affecting the evaluation of the potential of a test formulation to be bioequivalent to a reference formulation on the rate of drug absorption, the authors have proposed alternative approaches to the average bioequivalence methodology that is generally applied to pilot studies to overcome and reduce the uncertainty and to help pharmaceutical companies on the decision to go forward to pivotal bioequivalence studies. The G_mean_ ƒ_2_ factor using a cut-off of 35 was found to be most appropriate method in the simulation conditions frame, enabling them to more accurately conclude on the potential of test formulations, with a reduced sample size. For simplification, a decision tree is also proposed for an appropriate planning of the sample size and subsequent analysis approach to be followed in pilot BA/BE trials.

## Figures and Tables

**Figure 1 pharmaceutics-15-01430-f001:**
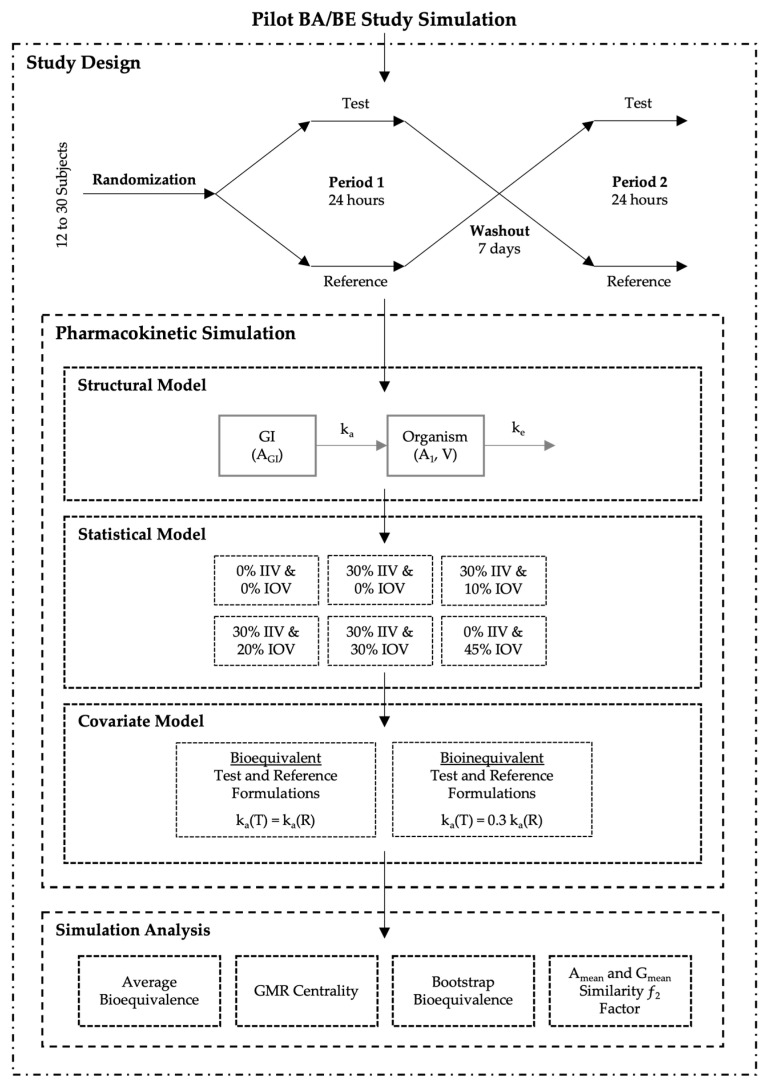
Pilot bioavailability/bioequivalence (BA/BE) trials simulation scheme.

**Figure 2 pharmaceutics-15-01430-f002:**
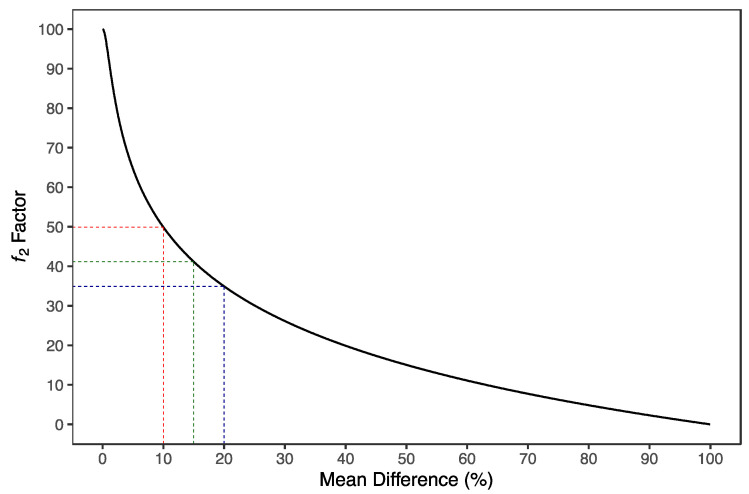
Distribution of ƒ_2_ similarity factor as a function of mean difference. ƒ_2_ similarity factor is derived from the mean squared difference and can be calculated as a function of the reciprocal of the mean squared-root transformation of the sum of square differences at all points. An average difference of 10%, 15%, and 20% from all measured time points results in a ƒ_2_ value of 50 (red dotted lines), 41 (green dotted lines) and 35 (blue dotted lines), respectively.

**Figure 3 pharmaceutics-15-01430-f003:**
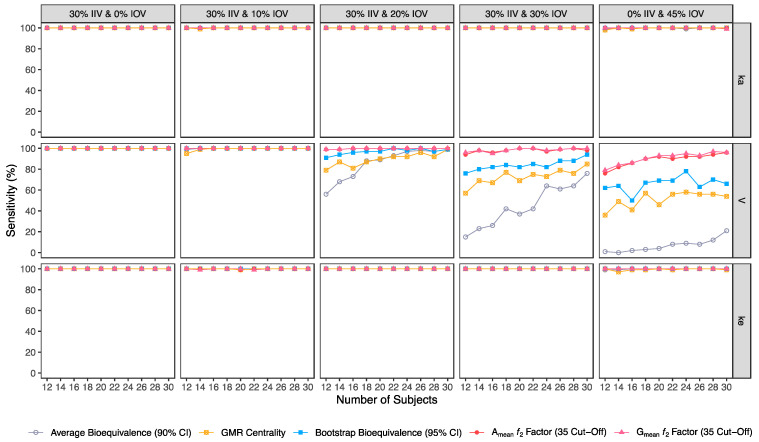
Variation of sensitivity for the bioequivalence evaluation methods (average bioequivalence, centrality of the test-to-reference GMR, bootstrap bioequivalence analysis, and A_mean_ and G_mean_ ƒ_2_ factor evaluated with a cut-off of 35) as function of the number of subjects, per tested variability for the different pharmacokinetic model parameters.

**Figure 4 pharmaceutics-15-01430-f004:**
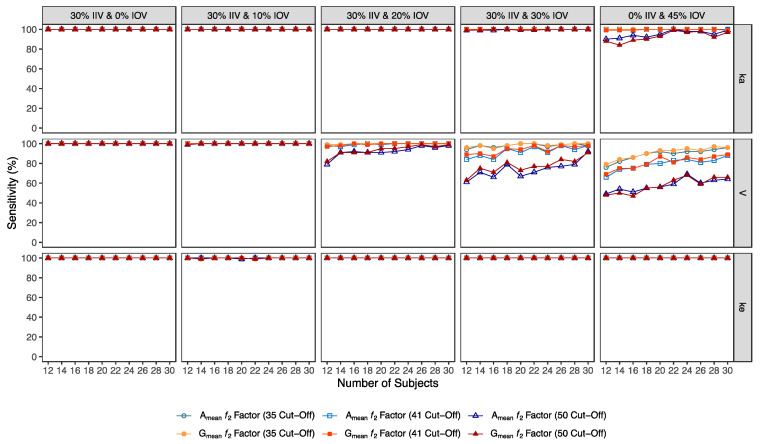
Variation of sensitivity for A_mean_ and G_mean_ ƒ_2_ factor evaluated with a cut-off of 35, 41, and 50, as function of the number of subjects, per tested variability for the different pharmacokinetic model parameters.

**Figure 5 pharmaceutics-15-01430-f005:**
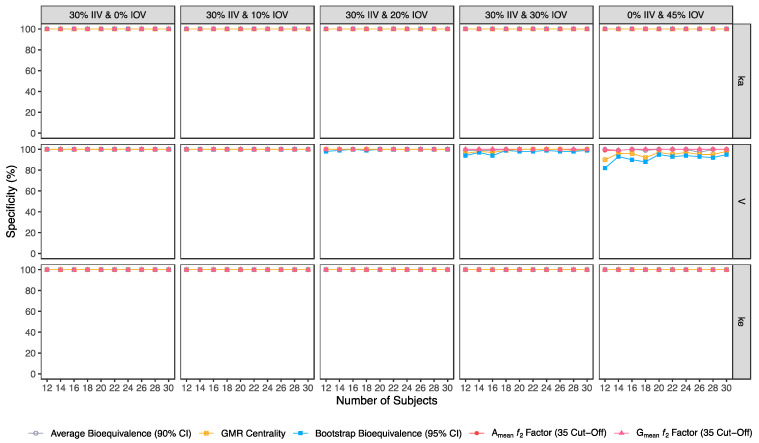
Variation of specificity for the bioequivalence evaluation methods (average bioequivalence, centrality of the test-to-reference GMR, bootstrap bioequivalence analysis, and A_mean_ and G_mean_ ƒ_2_ factor evaluated with a cut-off of 35) as function of the number of subjects, per tested variability for the different pharmacokinetic model parameters.

**Figure 6 pharmaceutics-15-01430-f006:**
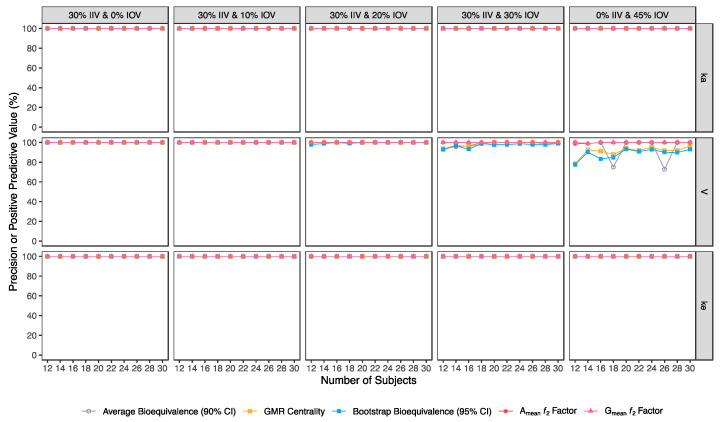
Variation of precision for the bioequivalence evaluation methods (average bioequivalence, centrality of the test-to-reference GMR, bootstrap bioequivalence analysis, and A_mean_ and G_mean_ ƒ_2_ factor evaluated with a cut-off of 35) as function of the number of subjects, per tested variability for the different pharmacokinetic model parameters.

**Figure 7 pharmaceutics-15-01430-f007:**
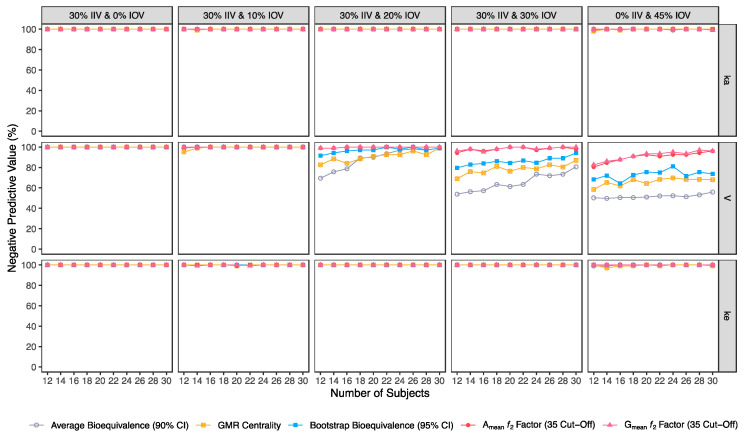
Variation of negative predictive value (NPV) for the bioequivalence evaluation methods (average bioequivalence, centrality of the test-to-reference GMR, bootstrap bioequivalence analysis, and A_mean_ and G_mean_ ƒ_2_ factor evaluated with a cut-off of 35) as function of the number of subjects, per tested variability for the different pharmacokinetic model parameters.

**Figure 8 pharmaceutics-15-01430-f008:**
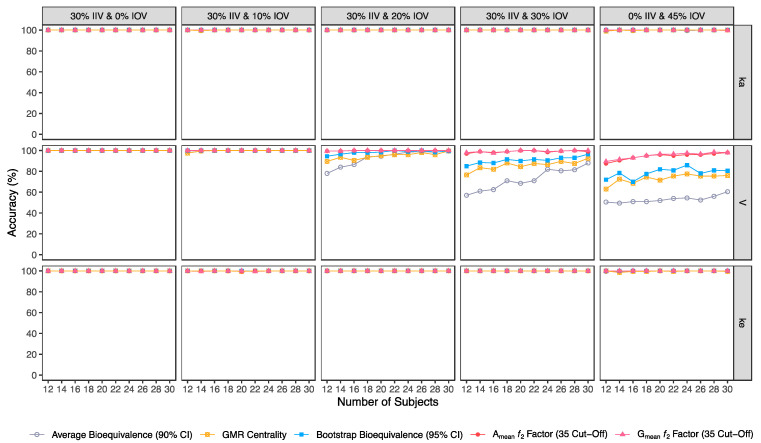
Variation of accuracy for the bioequivalence evaluation methods (average bioequivalence, centrality of the test-to-reference GMR, bootstrap bioequivalence analysis, and A_mean_ and G_mean_ ƒ_2_ factor evaluated with a cut-off of 35) as a function of the number of subjects, per tested variability for the different pharmacokinetic model parameters.

**Figure 9 pharmaceutics-15-01430-f009:**
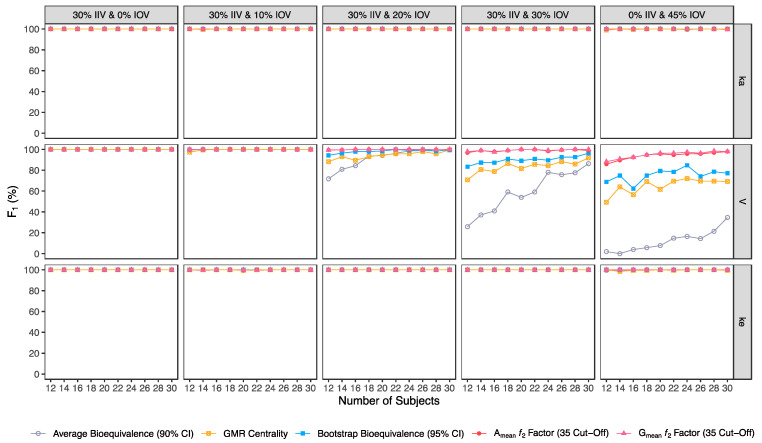
Variation of F_1_ for the bioequivalence evaluation methods (average bioequivalence, centrality of the test-to-reference GMR, bootstrap bioequivalence analysis, and A_mean_ and G_mean_ ƒ_2_ factor evaluated with a cut-off of 35) as a function of the number of subjects, per tested variability for the different pharmacokinetic model parameters.

**Figure 10 pharmaceutics-15-01430-f010:**
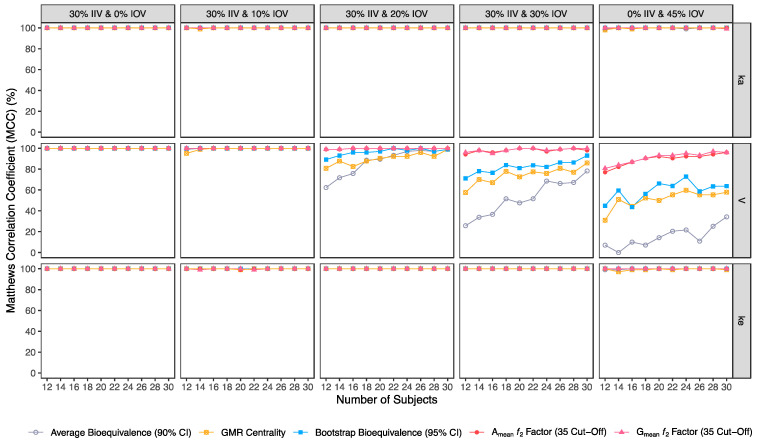
Variation of Matthews correlation coefficient (MCC) for the bioequivalence evaluation methods (average bioequivalence, centrality of the test-to-reference GMR, bootstrap bioequivalence analysis, and A_mean_ and G_mean_ ƒ_2_ factor evaluated with a cut-off of 35) as function of the number of subjects, per tested variability for the different pharmacokinetic model parameters.

**Figure 11 pharmaceutics-15-01430-f011:**
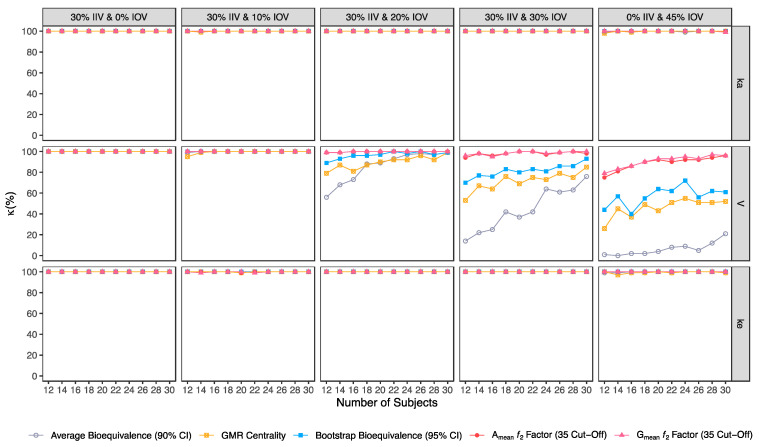
Variation of Cohen’s κ for the bioequivalence evaluation methods (average bioequivalence, centrality of the test-to-reference GMR, bootstrap bioequivalence analysis, and A_mean_ and G_mean_ ƒ_2_ factor evaluated with a cut-off of 35) as a function of the number of subjects, per tested variability for the different pharmacokinetic model parameters.

**Figure 12 pharmaceutics-15-01430-f012:**
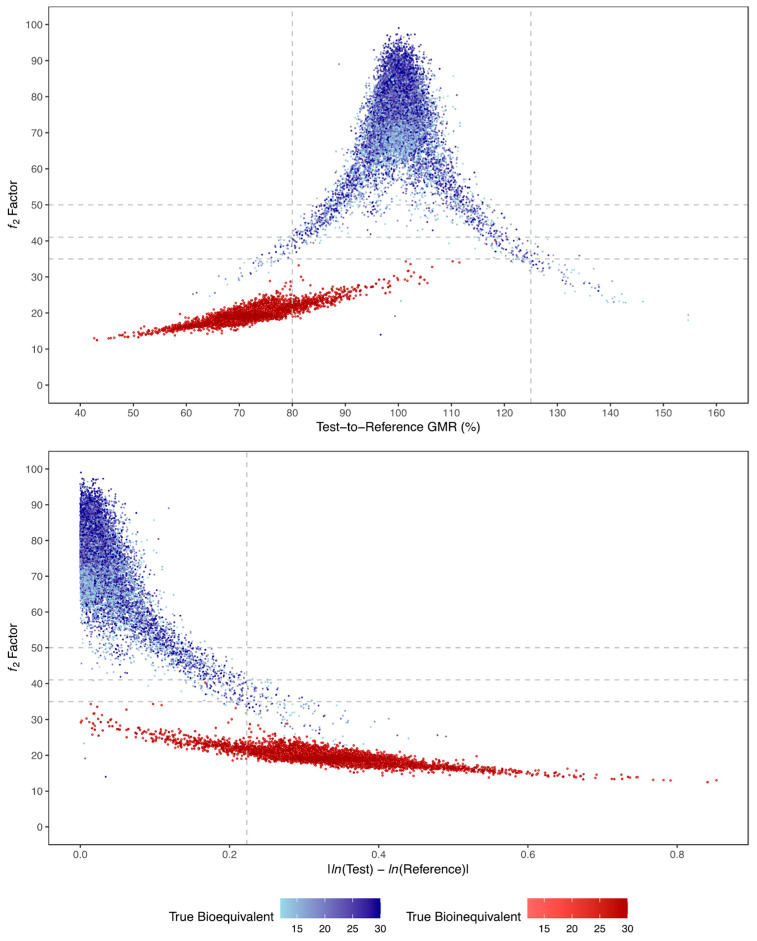
Relationship between G_mean_ *f*_2_ factor and test-to-reference GMR (above) or absolute LSM difference (below), and number of subjects (colour gradient), for all simulated true bioequivalent (blue) and true bioinequivalent (red) studies. Vertical dotted lines correspond to the maximum 20% difference between test and reference formulations, tested by the average bioequivalence approach. Horizontal dotted lines correspond to the tested cut-off values for ƒ_2_ of 50, 41, and 35.

**Figure 13 pharmaceutics-15-01430-f013:**
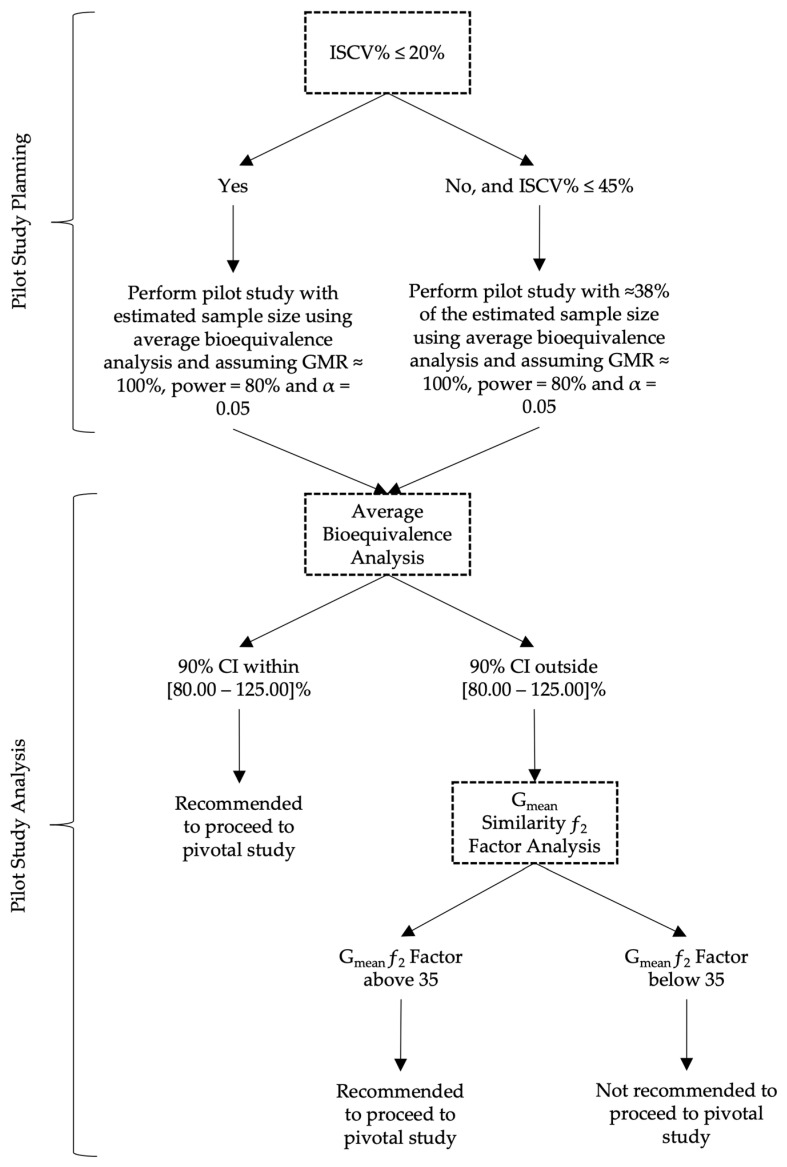
Proposed decision tree for the planning and analysis of pilot BA/BE studies.

**Table 1 pharmaceutics-15-01430-t001:** Compartmental Pharmacokinetic Parameters Initial Estimates.

k_a_ (h^−1^)	V (L)	k_e_ (h^−1^)	F
1.22	58.8	0.150	0.900

F: Absolute bioavailability, k_a_: absorption rate constant, k_e_: elimination rate constant, V: volume of distribution.

**Table 2 pharmaceutics-15-01430-t002:** Confusion Matrix of the Observed and Predicted Classes with Associated Statistics.

		Method Prediction		
		Bioequivalent	Bioinequivalent	
**Truly**	**Bioequivalent**	TP	FN Type II Error	Sensitivity $\frac{\mathrm{TP}}{\mathrm{TP}\;\;+\mathrm{FN}}$
	**Bioinequivalent**	FP Type I Error	TN	Specificity $\frac{\mathrm{TN}}{\mathrm{TN}\;+\;\mathrm{FP}}$
		Precision $\frac{\mathrm{TP}}{\mathrm{TP}\;+\;\mathrm{FP}}$	Negative Predictive Value $\frac{\mathrm{TN}}{\mathrm{TN}\;+\;\mathrm{FN}}$	Accuracy $\frac{\mathrm{TP}\;+\;\mathrm{TN}}{\mathrm{TP}\;+\;\mathrm{TN}\;+\;\mathrm{FP}\;+\;\mathrm{FN}}$

FN—False Negatives; FP—False Positives, TN—True Negatives, TP—True Positives.

**Table 3 pharmaceutics-15-01430-t003:** Cross-Tabulated Matrix Statistics Calculated for Average Bioequivalence (90% CI).

	Average Bioequivalence (90% CI)									
	Sensitivity (%)	Type II Error (%)	Specificity (%)	Type I Error (%)	Precision (%)	NPV (%)	Accuracy (%)	F_1_ (%)	MCC (%)	κ (%)
Baseline	100	0.00	100	0.00	100	100	100	100	100	100
k_a_										
30% IIV & 0% IOV	100	0.00	100	0.00	100	100	100	100	100	100
30% IIV & 10% IOV	100	0.00	100	0.00	100	100	100	100	100	100
30% IIV & 20% IOV	100	0.00	100	0.00	100	100	100	100	100	100
30% IIV & 30% IOV	100	0.00	100	0.00	100	100	100	100	100	100
0% IIV & 45% IOV	99.0–100	1.00–0.00	100	0.00	100	99.0–100	99.5–100	99.5–100	99.0–100.0	99.0–100
V										
30% IIV & 0% IOV	100	0.00	100	0.00	100	100	100	100	100	100
30% IIV & 10% IOV	99.0–100	1.00–0.00	100	0.00	100	99.0–100	99.5–100	99.5–100	99.0–100	99.0–100
30% IIV & 20% IOV	56.0–99.0	44.0–1.00	100	0.00	100	69.4–99.0	78.0–99.5	71.8–99.5	62.4–99.0	56.0–99.0
30% IIV & 30% IOV	15.0–76.0	85.0–24.0	99.0–100	1.00–0.00	93.75–100	53.8–80.6	57.0–88.0	25.9–86.4	25.8–78.3	14.0–76.0
0% IIV & 45% IOV	1.00–21.0	99.0–79.0	100	0.00	100	50.3–55.9	50.5–60.5	1.98–34.7	7.09–34.3	1.00–21.0
k_e_										
30% IIV & 0% IOV	100	0.00	100	0.00	100	100	100	100	100	100
30% IIV & 10% IOV	100	0.00	100	0.00	100	100	100	100	100	100
30% IIV & 20% IOV	100	0.00	100	0.00	100	100	100	100	100	100
30% IIV & 30% IOV	100	0.00	100	0.00	100	100	100	100	100	100
0% IIV & 45% IOV	99.0–100	1.00–0.00	100	0.00	100	99.0–100	99.5–100	99.5–100	99.0–100	99.0–100

Values represent the range calculated from simulated studies with 12 and 30 subjects. When statistics do not change between 12 and 30 subjects, unique values are presented instead of ranges. F_1_—Harmonic mean of sensitivity and precision; κ—Cohen’s Kappa; MCC—Matthews correlation coefficient; NPV—Negative predictive value.

**Table 4 pharmaceutics-15-01430-t004:** Cross-Tabulated Matrix Statistics Calculated for test-to-reference GMR Centrality.

	Test-to-Reference GMR Centrality									
	Sensitivity (%)	Type II Error (%)	Specificity (%)	Type I Error (%)	Precision (%)	NPV (%)	Accuracy (%)	F_1_ (%)	MCC (%)	κ (%)
Baseline	100	0.00	100	0.00	100	100	100	100	100	100
k_a_										
30% IIV & 0% IOV	100	0.00	100	0.00	100	100	100	100	100	100
30% IIV & 10% IOV	100	0.00	100	0.00	100	100	100	100	100	100
30% IIV & 20% IOV	100	0.00	100	0.00	100	100	100	100	100	100
30% IIV & 30% IOV	100	0.00	100	0.00	100	100	100	100	100	100
0% IIV & 45% IOV	98.0–100	2.00–0.00	100	0.00	100	98.0–100	99.0–100	99.0–100	98.02–100	98.0–100
V										
30% IIV & 0% IOV	100	0.00	100	0.00	100	100	100	100	100	100
30% IIV & 10% IOV	95.0–100	5.00–0.00	100	0.00	100	95.2–100	97.5–100	97.4–100	95.1–100	95.0–100
30% IIV & 20% IOV	79.0–99.0	21.0–1.00	100	0.00	100	82.6–99.0	89.5–99.5	88.3–99.5	80.8–99.0	79.0–99.0
30% IIV & 30% IOV	57.0–85.0	43.0–15.0	96.0–100	4.00–0.00	93.4–100	69.1–87.0	76.5–92.5	70.8–91.9	57.6–86.0	53.0–85.0
0% IIV & 45% IOV	36.0–54.0	64.0–46.0	90.0–98.0	10.0–2.00	78.3–96.4	58.4–68.1	63.0–76.0	49.3–69.2	30.9–57.9	26.0–52.0
k_e_										
30% IIV & 0% IOV	100	0.00	100	0.00	100	100	100	100	100	100
30% IIV & 10% IOV	100	0.00	100	0.00	100	100	100	100	100	100
30% IIV & 20% IOV	100	0.00	100	0.00	100	100	100	100	100	100
30% IIV & 30% IOV	100	0.00	100	0.00	100	100	100	100	100	100
0% IIV & 45% IOV	99.0–100	1.00–0.00	100	0.00	100	99.0–100	99.5–100	99.5–100	99.0–100	99.0–100

Values represent the range calculated from simulated studies with 12 and 30 subjects. When statistics do not change between 12 and 30 subjects, unique values are presented instead of ranges. F_1_—Harmonic mean of sensitivity and precision; κ—Cohen’s Kappa; MCC—Matthews correlation coefficient; NPV—Negative predictive value.

**Table 5 pharmaceutics-15-01430-t005:** Cross-Tabulated Matrix Statistics Calculated for Bootstrap Bioequivalence (95% CI).

	Bootstrap Bioequivalence (95% CI)									
	Sensitivity (%)	Type II Error (%)	Specificity (%)	Type I Error (%)	Precision (%)	NPV (%)	Accuracy (%)	F_1_ (%)	MCC (%)	κ (%)
Baseline	100	0.00	100	0.00	100	100	100	100	100	100
k_a_										
30% IIV & 0% IOV	100	0.00	100	0.00	100	100	100	100	100	100
30% IIV & 10% IOV	100	0.00	100	0.00	100	100	100	100	100	100
30% IIV & 20% IOV	100	0.00	100	0.00	100	100	100	100	100	100
30% IIV & 30% IOV	100	0.00	100	0.00	100	100	100	100	100	100
0% IIV & 45% IOV	100	0.00	100	0.00	100	100	100	100	100	100
V										
30% IIV & 0% IOV	100	0.00	100	0.00	100	100	100	100	100	100
30% IIV & 10% IOV	100	0.00	100	0.00	100	100	100	100	100	100
30% IIV & 20% IOV	91.0–99.0	9.00–1.00	98.0–100	2.00–0.00	97.85–100	91.6–99.0	94.5–99.5	94.3–99.5	89.2–99.0	89.0–99.0
30% IIV & 30% IOV	76.0–94.0	24.0–6.00	94.0–99.0	6.00–1.00	92.7–98.9	79.7–94.3	85.0–96.5	83.5–96.4	71.2–93.1	70.0–93.0
0% IIV & 45% IOV	62.0–66.0	38.0–34.0	82.0–95.0	18.0–5.00	77.5–93.0	68.3–73.6	72.0–80.5	68.9–77.2	44.9–63.7	44.0–61.0
k_e_										
30% IIV & 0% IOV	100	0.00	100	0.00	100	100	100	100	100	100
30% IIV & 10% IOV	100	0.00	100	0.00	100	100	100	100	100	100
30% IIV & 20% IOV	100	0.00	100	0.00	100	100	100	100	100	100
30% IIV & 30% IOV	100	0.00	100	0.00	100	100	100	100	100	100
0% IIV & 45% IOV	100	0.00	100	0.00	100	100	100	100	100	100

Values represent the range calculated from simulated studies with 12 and 30 subjects. When statistics do not change between 12 and 30 subjects, unique values are presented instead of ranges. F_1_—Harmonic mean of sensitivity and precision; κ—Cohen’s Kappa; MCC—Matthews correlation coefficient; NPV—Negative predictive value.

**Table 6 pharmaceutics-15-01430-t006:** Cross-Tabulated Matrix Statistics Calculated for Arithmetic Mean (A_mean_) ƒ_2_ Factor, Using a Cut-Off of 35.

	A_mean_ ƒ_2_ Factor (Cut-Off of 35)									
	Sensitivity (%)	Type II Error (%)	Specificity (%)	Type I Error (%)	Precision (%)	NPV (%)	Accuracy (%)	F_1_ (%)	MCC (%)	κ (%)
Baseline	100	0.00	100	0.00	100	100	100	100	100	100
k_a_										
30% IIV & 0% IOV	100	0.00	100	0.00	100	100	100	100	100	100
30% IIV & 10% IOV	100	0.00	100	0.00	100	100	100	100	100	100
30% IIV & 20% IOV	100	0.00	100	0.00	100	100	100	100	100	100
30% IIV & 30% IOV	100	0.00	100	0.00	100	100	100	100	100	100
0% IIV & 45% IOV	100	0.00	100	0.00	100	100	100	100	100	100
V										
30% IIV & 0% IOV	100	0.00	100	0.00	100	100	100	100	100	100
30% IIV & 10% IOV	100	0.00	100	0.00	100	100	100	100	100	100
30% IIV & 20% IOV	99.0–100	1.00–0.00	100	0.00	100	99.0–100	99.5–100	99.5–100	99.0–100	99.0–100
30% IIV & 30% IOV	94.0–98.0	6.00–2.00	100	0.00	100	94.3–98.0	97.0–99.0	96.9–99.0	94.2–98.0	94.0–98.0
0% IIV & 45% IOV	76.0–96.0	24.0–4.00	99.0–100	1.00–0.00	98.7–100	80.5–96.2	87.5–98.0	85.9–98.0	77.1–96.1	75.0–96.0
k_e_										
30% IIV & 0% IOV	100	0.00	100	0.00	100	100	100	100	100	100
30% IIV & 10% IOV	100	0.00	100	0.00	100	100	100	100	100	100
30% IIV & 20% IOV	100	0.00	100	0.00	100	100	100	100	100	100
30% IIV & 30% IOV	100	0.00	100	0.00	100	100	100	100	100	100
0% IIV & 45% IOV	100	0.00	100	0.00	100	100	100	100	100	100

Values represent the range calculated from simulated studies with 12 and 30 subjects. When statistics do not change between 12 and 30 subjects, unique values are presented instead of ranges. F_1_—Harmonic mean of sensitivity and precision; κ—Cohen’s Kappa; MCC—Matthews correlation coefficient; NPV—Negative predictive value.

**Table 7 pharmaceutics-15-01430-t007:** Cross-Tabulated Matrix Statistics Calculated for Geometric Mean (G_mean_) ƒ_2_ Factor, Using a Cut-Off of 35.

	G_mean_ ƒ_2_ Factor (Cut-Off of 35)									
	Sensitivity (%)	Type II Error (%)	Specificity (%)	Type I Error (%)	Precision (%)	NPV (%)	Accuracy (%)	F_1_ (%)	MCC (%)	κ (%)
Baseline	100	0.00	100	0.00	100	100	100	100	100	100
k_a_										
30% IIV & 0% IOV	100	0.00	100	0.00	100	100	100	100	100	100
30% IIV & 10% IOV	100	0.00	100	0.00	100	100	100	100	100	100
30% IIV & 20% IOV	100	0.00	100	0.00	100	100	100	100	100	100
30% IIV & 30% IOV	100	0.00	100	0.00	100	100	100	100	100	100
0% IIV & 45% IOV	100	0.00	100	0.00	100	100	100	100	100	100
V										
30% IIV & 0% IOV	100	0.00	100	0.00	100	100	100	100	100	100
30% IIV & 10% IOV	100	0.00	100	0.00	100	100	100	100	100	100
30% IIV & 20% IOV	99.0–100	1.00–0.00	100	0.00	100	99.0–100	99.5–100	99.5–100	99.0–100	99.0–100
30% IIV & 30% IOV	96.0–100	4.00–0.00	100	0.00	100	96.2–100	98.0–100	98.0–100	96.1–100	96.0–100
0% IIV & 45% IOV	79.0–96.0	21.0–4.00	100	0.00	100	82.6–96.2	89.5–98.0	88.3–98.0	80.8–96.1	79.0–96.0
k_e_										
30% IIV & 0% IOV	100	0.00	100	0.00	100	100	100	100	100	100
30% IIV & 10% IOV	100	0.00	100	0.00	100	100	100	100	100	100
30% IIV & 20% IOV	100	0.00	100	0.00	100	100	100	100	100	100
30% IIV & 30% IOV	100	0.00	100	0.00	100	100	100	100	100	100
0% IIV & 45% IOV	100	0.00	100	0.00	100	100	100	100	100	100

Values represent the range calculated from simulated studies with 12 and 30 subjects. When statistics do not change between 12 and 30 subjects, unique values are presented instead of ranges. F_1_—Harmonic mean of sensitivity and precision; κ—Cohen’s Kappa; MCC—Matthews correlation Coefficient; NPV—Negative predictive value.

**Table 8 pharmaceutics-15-01430-t008:** Cross-Tabulated Matrix Statistics Calculated for Arithmetic Mean (A_mean_) ƒ_2_ Factor, Using a Cut-Off of 41.

	A_mean_ ƒ_2_ Factor (Cut-Off of 41)									
	Sensitivity (%)	Type II Error (%)	Specificity (%)	Type I Error (%)	Precision (%)	NPV (%)	Accuracy (%)	F_1_ (%)	MCC (%)	κ (%)
Baseline	100	0.00	100	0.00	100	100	100	100	100	100
k_a_										
30% IIV & 0% IOV	100	0.00	100	0.00	100	100	100	100	100	100
30% IIV & 10% IOV	100	0.00	100	0.00	100	100	100	100	100	100
30% IIV & 20% IOV	100	0.00	100	0.00	100	100	100	100	100	100
30% IIV & 30% IOV	100	0.00	100	0.00	100	100	100	100	100	100
0% IIV & 45% IOV	100	0.00	100	0.00	100	100	100	100	100	100
V										
30% IIV & 0% IOV	100	0.00	100	0.00	100	100	100	100	100	100
30% IIV & 10% IOV	100	0.00	100	0.00	100	100	100	100	100	100
30% IIV & 20% IOV	98.0–100	2.00–0.00	100	0.00	100	98.0–100	99.0–100	99.0–100	98.0–100	98.0–100
30% IIV & 30% IOV	84.0–98.0	16.0–2.00	100	0.00	100	86.2–98.0	92.0–99.0	91.3–99.0	85.1–98.0	84.0–98.0
0% IIV & 45% IOV	66.0–88.0	34.0–12.0	100	0.00	100	74.6–89.3	83.0–94.0	79.5–93.6	70.2–88.6	66.0–88.0
k_e_										
30% IIV & 0% IOV	100	0.00	100	0.00	100	100	100	100	100	100
30% IIV & 10% IOV	100	0.00	100	0.00	100	100	100	100	100	100
30% IIV & 20% IOV	100	0.00	100	0.00	100	100	100	100	100	100
30% IIV & 30% IOV	100	0.00	100	0.00	100	100	100	100	100	100
0% IIV & 45% IOV	100	0.00	100	0.00	100	100	100	100	100	100

Values represent the range calculated from simulated studies with 12 and 30 subjects. When statistics do not change between 12 and 30 subjects, unique values are presented instead of ranges. F_1_—Harmonic mean of sensitivity and precision; κ—Cohen’s Kappa; MCC—Matthews correlation coefficient; NPV—Negative predictive value.

**Table 9 pharmaceutics-15-01430-t009:** Cross-Tabulated Matrix Statistics Calculated for Geometric Mean (G_mean_) ƒ_2_ Factor, Using a Cut-Off of 41.

	G_mean_ ƒ_2_ Factor (Cut-Off of 41)									
	Sensitivity (%)	Type II Error (%)	Specificity (%)	Type I Error (%)	Precision (%)	NPV (%)	Accuracy (%)	F_1_ (%)	MCC (%)	κ (%)
Baseline	100	0.00	100	0.00	100	100	100	100	100	100
k_a_										
30% IIV & 0% IOV	100	0.00	100	0.00	100	100	100	100	100	100
30% IIV & 10% IOV	100	0.00	100	0.00	100	100	100	100	100	100
30% IIV & 20% IOV	100	0.00	100	0.00	100	100	100	100	100	100
30% IIV & 30% IOV	100	0.00	100	0.00	100	100	100	100	100	100
0% IIV & 45% IOV	99.0	1.00	100	0.00	100	99.0	99.5	99.5	99.0	99.0
V										
30% IIV & 0% IOV	100	0.00	100	0.00	100	100	100	100	100	100
30% IIV & 10% IOV	100	0.00	100	0.00	100	100	100	100	100	100
30% IIV & 20% IOV	97.0–100	3.00–0.00	100	0.00	100	97.1–100	98.5–100	98.5–100	97.0–100	97.0–100
30% IIV & 30% IOV	89.0–98.0	11.0–2.00	100	0.00	100	90.1–98.0	94.5–99.0	94.2–99.0	89.5–98.0	89.0–98.0
0% IIV & 45% IOV	69.0–89.0	31.0–11.0	100	0.00	100	76.3–90.1	84.5–94.5	81.7–94.2	72.6–89.5	69.0–89.0
k_e_										
30% IIV & 0% IOV	100	0.00	100	0.00	100	100	100	100	100	100
30% IIV & 10% IOV	100	0.00	100	0.00	100	100	100	100	100	100
30% IIV & 20% IOV	100	0.00	100	0.00	100	100	100	100	100	100
30% IIV & 30% IOV	100	0.00	100	0.00	100	100	100	100	100	100
0% IIV & 45% IOV	100	0.00	100	0.00	100	100	100	100	100	100

Values represent the range calculated from simulated studies with 12 and 30 subjects. When statistics do not change between 12 and 30 subjects, unique values are presented instead of ranges. F_1_—Harmonic mean of sensitivity and precision; κ—Cohen’s Kappa; MCC—Matthews correlation coefficient; NPV—Negative predictive value.

**Table 10 pharmaceutics-15-01430-t010:** Cross-Tabulated Matrix Statistics Calculated for Arithmetic Mean (A_mean_) ƒ_2_ Factor, Using a Cut-Off of 50.

	A_mean_ ƒ_2_ Factor (Cut-Off of 50)									
	Sensitivity (%)	Type II Error (%)	Specificity (%)	Type I Error (%)	Precision (%)	NPV (%)	Accuracy (%)	F_1_ (%)	MCC (%)	κ (%)
Baseline	100	0.00	100	0.00	100	100	100	100	100	100
k_a_										
30% IIV & 0% IOV	100	0.00	100	0.00	100	100	100	100	100	100
30% IIV & 10% IOV	100	0.00	100	0.00	100	100	100	100	100	100
30% IIV & 20% IOV	100	0.00	100	0.00	100	100	100	100	100	100
30% IIV & 30% IOV	99.0–100	1.00–0.00	100	0.00	100	99.0–100	99.5–100	99.5–100	99.0–100	99.0–100
0% IIV & 45% IOV	90.0–99.0	10.0–1.00	100	0.00	100	90.9–99.0	95.0–99.5	94.7–99.5	90.5–99.0	90.0–99.0
V										
30% IIV & 0% IOV	100	0.00	100	0.00	100	100	100	100	100	100
30% IIV & 10% IOV	99.0–100	1.00–0.00	100	0.00	100	99.0–100	99.5–100	99.5–100	99.0–100	99.0–100
30% IIV & 20% IOV	79.0–98.0	21.0–2.00	100	0.00	100	82.6–98.0	89.5–99.0	88.3–99.0	80.8–98.0	79.0–98.0
30% IIV & 30% IOV	61.0–92.0	39.0–8.00	100	0.00	100	71.9–92.6	80.5–96.0	75.8–95.8	66.3–92.3	61.0–92.0
0% IIV & 45% IOV	49.0–64.0	51.0–36.0	100	0.00	100	66.2–73.5	74.5–82.0	65.8–78.1	57.0–68.6	49.0–64.0
k_e_										
30% IIV & 0% IOV	100	0.00	100	0.00	100	100	100	100	100	100
30% IIV & 10% IOV	100	0.00	100	0.00	100	100	100	100	100	100
30% IIV & 20% IOV	100	0.00	100	0.00	100	100	100	100	100	100
30% IIV & 30% IOV	100	0.00	100	0.00	100	100	100	100	100	100
0% IIV & 45% IOV	100	0.00	100	0.00	100	100	100	100	100	100

Values represent the range calculated from simulated studies with 12 and 30 subjects. When statistics do not change between 12 and 30 subjects, unique values are presented instead of ranges. F_1_—Harmonic mean of sensitivity and precision; κ—Cohen’s Kappa; MCC—Matthews correlation coefficient; NPV—Negative predictive value.

**Table 11 pharmaceutics-15-01430-t011:** Cross-Tabulated Matrix Statistics Calculated for Geometric Mean (G_mean_) ƒ_2_ Factor, Using a Cut-Off of 50.

	G_mean_ ƒ_2_ Factor (Cut-Off of 50)									
	Sensitivity (%)	Type II Error (%)	Specificity (%)	Type I Error (%)	Precision (%)	NPV (%)	Accuracy (%)	F_1_ (%)	MCC (%)	κ (%)
Baseline	100	0.00	100	0.00	100	100	100	100	100	100
k_a_										
30% IIV & 0% IOV	100	0.00	100	0.00	100	100	100	100	100	100
30% IIV & 10% IOV	100	0.00	100	0.00	100	100	100	100	100	100
30% IIV & 20% IOV	100	0.00	100	0.00	100	100	100	100	100	100
30% IIV & 30% IOV	99.0–100	1.00–0.00	100	0.00	100	99.0–100	99.5–100	99.5–100	99.0–100	99.0–100
% IIV & 45% IOV	88.0–97.0	12.0–3.00	100	0.00	100	89.3–97.1	94.0–98.5	93.6–98.5	88.6–97.0	88.0–97.0
V										
30% IIV & 0% IOV	100	0.00	100	0.00	100	100	100	100	100	100
30% IIV & 10% IOV	99.0–100	1.00–0.00	100	0.00	100	99.0–100	99.5–100	99.5–100	99.0–100	99.0–100
30% IIV & 20% IOV	82.0–99.0	18.0–1.00	100	0.00	100	84.8–99.0	91.0–99.5	90.1–99.5	83.4–99.0	82.0–99.0
30% IIV & 30% IOV	63.0–91.0	37.0–9.00	100	0.00	100	73.0–91.7	81.5–95.5	77.3–95.3	67.8–91.4	63.0–91.0
0% IIV & 45% IOV	48.0–66.0	52.0–34.0	100	0.00	100	65.8–74.6	74.0–83.0	64.9–79.5	56.2–70.2	48.0–66.0
k_e_										
30% IIV & 0% IOV	100	0.00	100	0.00	100	100	100	100	100	100
30% IIV & 10% IOV	100	0.00	100	0.00	100	100	100	100	100	100
30% IIV & 20% IOV	100	0.00	100	0.00	100	100	100	100	100	100
30% IIV & 30% IOV	100	0.00	100	0.00	100	100	100	100	100	100
0% IIV & 45% IOV	100	0.00	100	0.00	100	100	100	100	100	100

Values represent the range calculated from simulated studies with 12 and 30 subjects. When statistics do not change between 12 and 30 subjects, unique values are presented instead of ranges. F_1_—Harmonic mean of sensitivity and precision; κ—Cohen’s Kappa; MCC—Matthews correlation coefficient; NPV—Negative predictive value.

**Table 12 pharmaceutics-15-01430-t012:** Sample Size for a 2 × 2 × 2 Crossover Study for Different Bioequivalence Evaluation Methods, Targeting a Power of at Least 80%, an α of 0.05, and Assuming a GMR of 100%.

IOV (%)	Average Bioequivalence ^1^	Bootstrap Bioequivalence	G_mean_ ƒ_2_ Factor		
			35	41	50
10%	12	12	12	12	12
20%	16	12	12	12	12
30%	32	14	12	12	18
45%	66	>30	14	20	>30

^1^ Calculated using R package ‘PowerTOST’ version 1.5–3 [14]. G_mean_—Geometric mean; IOV—Inter-occasion variability.

## Data Availability

Not applicable.

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
