# Peer review of "Alternative Analysis Approaches for the Assessment of Pilot Bioavailability/Bioequivalence Studies"

_pharmaceutics, 2023, doi:10.3390/pharmaceutics15051430_

Round 1

Reviewer 1 Report

This paper is a good work with a high significance of content due to this is  a way to overcome and reduce the uncertainty and to help pharmaceutical companies on the decision to go forward to pivotal bioequivalence studies.

Just some minor comments to take into account:

Why IIV is 30% as maximun variability? High variability is consider more than 30%.

References are not well conected

Author Response

Comment:

Why IIV is 30% as maximum variability? High variability is consider more than 30%.

Response:

The authors acknowledge Reviewer’s comments.

Regarding inter-individual variability (IIV), a maximum of 30% was used in 4 of the 6 variability scenarios tested (see lines 124-126 of manuscript). IIV is not expected to impact the outcome of the analysis in a crossover bioequivalence study, as in a crossover design each subject is administered with both Test and Reference products, therefore acting as its own control. IIV is consequently suppressed.

Such a lack of impact is demonstrated by the lack of differences between the different methods results (ABE and alternative) for the baseline simulations (0% IIV and 0% inter-occasion variability [IOV]) and the results for the simulations with 30% IIV and 0% IOV. Consequently, considering that IIV would not interfere in the observed results, the authors decided not to test different levels of IIV other than 0% and 30%.

Comment:

References are not well connected.

Response:

Regarding bibliographic references, the authors made a revision in the text. According to the editor, hyperlinks will be added to the final version.

Reviewer 2 Report

General comments:

In my opinion, this is an interesting paper, which would provide valuable insights in decision-making, mainly inside the companies. However (precisely because of this) the approach, the conclusions and the discussion should be better explained and nuanced.

Its most interesting and novel aspect is the design of the simulation study. Usually, similar studies are based on the direct generation of individual observations in a crossover design (i.e., assuming as true assumptions such as the normality of log-transformed bioavailability variables) or, even more directly, on the generation of relevant statistics like the within-subject sample variance (e.g., according to a chi-square distribution). The authors propose much more realistic simulations, closer to what actually happens in practice.

I believe that the main shortcoming of this article is the failure to consider two-stage designs, which are also taken into account by the regulations. In these designs, the pilot study would correspond to the first stage. Admittedly, in an implicit way in the conclusions, Figure 13 suggests a two-stage design. However, there is a lot of literature on two-stage designs in bioequivalence, where important subjects like the type I error control are considered.

More specific comments and suggestions:

In agreement with the above general comment, under the general perspective of two-stage designs, (line 74) proceeding further with a pivotal study in an scenario of non-bioequivalence does not necessarily imply a type I error –the situation can already be corrected in the second stage, which here corresponds to the pivotal study.

The bootstrap approach should be better explained, and possibly improved. In order to make the bootstrap better mimic the real experiments, typically the sample size of each bootstrap resample is taken as equal to the sample size of the “true data” (i.e., “n = n1 + n2” “true but simulated” concentration-time profiles, n1 corresponding to the first sequence of the crossover and n2 to the second one). From line 202, a bootstrap sample size inflation seems to be proposed, in order to achieve a given power in the bootstrap data, but representing worse the initially simulated “true” pharmacokinetic data. Another potentially weak point of the bootstrap approach is the use of the basic percentile bootstrap confidence interval. There are much better bootstrap confidence intervals. I guess that the observed excess of type I errors (section 3.2.3) may be associated to almost one of the above points.

The observed average bioequivalence (ABE) method lower accuracy (e.g., section 3.2.5) might be due to the artifact that not entirely comparable things are being compared. ABE falls entirely within the classical paradigm of hypothesis testing, i.e., controlling type I error (here the user’s risk, declaring as bioequivalent a non-bioequivalent product) even at the expense of power. As they are applied, this constraint does not exist for the other methods in which accuracy (i.e., jointly considering the correct decisions of declaring bioequivalence when it holds and not declaring it in non-bioequivalence scenarios, TP + TN) makes more sense. On the other hand, while valuable for consideration, these other approaches will hardly be taken into account in regulations.

In the discussion and concluding remarks, I observe some overuse of the term “significant”. In fact, what the authors are discussing to compare methods are some performance descriptive measures (sensitivity, accuracy…), which may show more or less “important” differences but not necessarily “(statistically) significant” because no inferential analysis is performed on them.

In conclusion, I suggest acceptance after a major revision, mainly from section 3.2.5 and the following sections. I also believe that considering some of the other suggestions might improve this paper interest, especially in a special issue devoted to these topics.

Author Response

Comment:

I believe that the main shortcoming of this article is the failure to consider two-stage designs, which are also taken into account by the regulations. In these designs, the pilot study would correspond to the first stage. Admittedly, in an implicit way in the conclusions, Figure 13 suggests a two-stage design. However, there is a lot of literature on two-stage designs in bioequivalence, where important subjects like the type I error control are considered.

Response:

The authors acknowledge Reviewer’s comment.

A two-stage design approach in bioequivalence studies has been allowed by the European Medicines Agency (EMA) since 2010. Despite proposals for the use of the two-stage procedure in bioequivalence studies appeared in the 80s, and the papers of Potvin et al. [1] and then of Montague et al. [2] became the starting point for consideration of the use of the two-stage design in bioequivalence studies, generic pharmaceutical companies keep sceptic on its application. One reason seems to be the lack of practical guidance for Sponsors on when and how the two-stage design can be beneficial in bioequivalence studies. Another reason might also be the advanced statistical simulations and arguments which may give the impression that the two-stage study is complicated and risky [3].

In the present work, authors have opted to follow the current status on drug product development and did not consider the interesting option to follow a two-stage approach. However, this work aims firstly to explore alternative approaches to the average bioequivalence methodology that is generally applied to pilot studies. The potential use of the developed methodology to be included in stopping rules in two stage designs should be further tested and subject to a new scientific paper.

References

[1] Potvin, D., DiLiberti, C.E., Hauck, W.W., Parr, A.F., Schuirmann, D.J. and Smith, R.A. (2008), Sequential design approaches for bioequivalence studies with crossover designs. Pharmaceut. Statist., 7: 245-262. https://doi.org/10.1002/pst.294

[2] Montague, T. H., Potvin, D., Diliberti, C. E., Hauck, W. W., Parr, A. F., & Schuirmann, D. J. (2012). Additional results for 'Sequential design approaches for bioequivalence studies with crossover designs'. Pharmaceutical statistics, 11(1), 8–13. https://doi.org/10.1002/pst.483

[3] Kaza, M., Sokolovskyi, A. & Rudzki, P.J. 10th Anniversary of a Two-Stage Design in Bioequivalence. Why Has it Still Not Been Implemented?. Pharm Res 37, 140 (2020). https://doi.org/10.1007/s11095-020-02871-3

Comment:

In agreement with the above general comment, under the general perspective of two-stage designs, (line 74) proceeding further with a pivotal study in a scenario of non-bioequivalence does not necessarily imply a type I error –the situation can already be corrected in the second stage, which here corresponds to the pivotal study.

Response:

The authors acknowledge Reviewer’s comment.

The meaning intended to the following sentence is that due variability, results from an unpowered sample size pilot study may lead to a wrong decision to move further with a pivotal study. In the case of truly bioinequivalent formulations, pilot study results may wrongly conclude for its bioequivalence, giving confidence to sponsors on the potentiality of Test formulation to show bioequivalence to the Reference formulation in a pivotal study. On the other hand, in the case of truly bioequivalent formulations, pilot study results may wrongly conclude for its bioinequivalence, giving no confidence to sponsors to perform a pivotal study. Therefore, at this stage of drug product development, the risk of the decision to move forward into a pivotal study relies entirely with the Sponsor.

«Consequently, there is a greater risk of either (i) validating a bioinequivalent Test formulation and proceed further with a pivotal study (falling into type I error) or (ii) discard a potentially bioequivalent formulation, by not conducting a pivotal study (falling into type II error).»

Therefore, the authors propose to reword this sentence, deleting “(falling into type I error)” and “(falling into type II error)”, as below:

«Consequently, there is a greater risk of either (i) validating a bioinequivalent Test formulation and proceed further with a pivotal study or (ii) discard a potentially bioequivalent formulation, by not conducting a pivotal study»

Comment:

The bootstrap approach should be better explained, and possibly improved. In order to make the bootstrap better mimic the real experiments, typically the sample size of each bootstrap resample is taken as equal to the sample size of the “true data” (i.e., “n = n1 + n2” “true but simulated” concentration-time profiles, n1 corresponding to the first sequence of the crossover and n2 to the second one). From line 202, a bootstrap sample size inflation seems to be proposed, in order to achieve a given power in the bootstrap data, but representing worse the initially simulated “true” pharmacokinetic data. Another potentially weak point of the bootstrap approach is the use of the basic percentile bootstrap confidence interval. There are much better bootstrap confidence intervals. I guess that the observed excess of type I errors (section 3.2.3) may be associated to almost one of the above points.

Response:

The authors acknowledge Reviewer’s comment.

The aim of the bootstrap methodology was to aid Sponsors to predict the outcome of a pivotal study, designed from the performed pilot study. Therefore, the number of subjects in each resample corresponds to the sample size necessary to reach a power of at least 80%, considering the variability found in the each simulated pilot study, i.e., the intra-subject coefficient of variation (ISCV%).

Regarding the choice of the calculation of the confidence intervals for the bootstrap methodology, the authors are aware of the existence of other approaches such as the bias-corrected and accelerated (BCa) bootstrap interval, which adjust for skewness in the bootstrap distribution, making it more sophisticated than the simpler percentiles’ method. The choice of the simpler percentiles’ method relied mainly in optimizing and significantly reduce the run time of the of the entire project, considering 32 000 trials were simulated. Moreover, since simulated data showed a log-normal distribution (Appendix), skewness of data was not pondered.

Comment:

The observed average bioequivalence (ABE) method lower accuracy (e.g., section 3.2.5) might be due to the artifact that not entirely comparable things are being compared. ABE falls entirely within the classical paradigm of hypothesis testing, i.e., controlling type I error (here the user’s risk, declaring as bioequivalent a non-bioequivalent product) even at the expense of power. As they are applied, this constraint does not exist for the other methods in which accuracy (i.e., jointly considering the correct decisions of declaring bioequivalence when it holds and not declaring it in non-bioequivalence scenarios, TP + TN) makes more sense. On the other hand, while valuable for consideration, these other approaches will hardly be taken into account in regulations.

Response:

The authors acknowledge Reviewer’s comment and totally agree with it.

Comment:

In the discussion and concluding remarks, I observe some overuse of the term “significant”. In fact, what the authors are discussing to compare methods are some performance descriptive measures (sensitivity, accuracy…), which may show more or less “important” differences but not necessarily “(statistically) significant” because no inferential analysis is performed on them.

Response:

The authors acknowledge Reviewer’s comment.

Term “significant” was revised along the entire document and was replaced by contextualized terms in lines 378, 426, 464, 656 and 661.

Reviewer 3 Report

This article describes alternative approaches to the average bioequivalence methodology that overcome and reduce the uncertainty on the conclusions of these studies and on the potential of test formulations. Overall, the manuscript is well written, and the results are logically and clearly presented. In my opinion, the paper can be published.

There are also a few minor typos that need addressing.

Author Response

Comment:

There are also a few minor typos that need addressing.

Response:

The authors acknowledge Reviewer’s comment.

The authors have reviewed the entire document for typos and will address corrections.